# Engineering brain activity patterns by neuromodulator polytherapy for treatment of disorders

Mostafa Ghannad-Rezaie [1,2], Peter M. Eimon[1], Yuelong Wu[1] & Mehmet Fatih Yanik[1,2]

Conventional drug screens and treatments often ignore the underlying complexity of brain network dysfunctions, resulting in suboptimal outcomes. Here we ask whether we can correct abnormal functional connectivity of the entire brain by identifying and combining multiple neuromodulators that perturb connectivity in complementary ways. Our approach avoids the combinatorial complexity of screening all drug combinations. We develop a high-speed platform capable of imaging more than 15000 neurons in 50ms to map the entire brain functional connectivity in large numbers of vertebrates under many conditions. Screening a panel of drugs in a zebrafish model of human Dravet syndrome, we show that even drugs with related mechanisms of action can modulate functional connectivity in significantly different ways. By clustering connectivity fingerprints, we algorithmically select small subsets of complementary drugs and rapidly identify combinations that are significantly more effective at correcting abnormal networks and reducing spontaneous seizures than monotherapies, while minimizing behavioral side effects. Even at low concentrations, our polytherapy performs superior to individual drugs even at highest tolerated concentrations.

[1] Massachusetts Institute of Technology, Cambridge, MA 02139, USA. [2] Institute for Neuroinformatics, ETH, Zurich 8092, Switzerland. Correspondence and requests for materials should be addressed to M.F.Y. (email: yanik@ethz.ch)

Treatment-resistant neurological and psychiatric disorders are a major public health problem impacting many millions of people worldwide. Existing therapeutics are ineffective in 40% of patients with anxiety disorders[1], 30–40% of patients with depression[2], and 20–30% of patients with epilepsy[3–5], 30% of patients with schizophrenia[6], and 40% of patients with OCD[7]. Although drug discovery often relies on identifying novel therapeutics with improved efficacy and fewer side effects, combining existing drugs (i.e. polytherapy) may provide better outcomes. Polytherapy utilizes two or more drugs, often with different mechanisms of action (MOAs), in order to achieve higher efficacy and reduced side effects. Polytherapy is already widely used in the clinical treatment of many disorders. For example, there is strong evidence for synergism when valproate and lamotrigine are used as a duotherapy for partial and generalized epilepsy[8,9]. A large-scale study of patients with bipolar disorder found that 33% were put on a polytherapy regimen (defined as two or more major psychotropic drugs) beginning with their initial prescription[10]. Even in the case of schizophrenia and related psychoses, where treatment guidelines recommend monotherapy, it is estimated that ~40–60% of patients wind up receiving antipsychotic polytherapy[11]. In addition, it is increasingly apparent that even many monotherapies can have more than one MOA at therapeutic concentrations. For example, there is convincing evidence that many modern antiepileptic agents—including felbamate, topiramate, levetiracetam, and zonisamide—work through multiple targets[12].

Unfortunately, empirical approaches for selecting the optimal polytherapy combinations are challenging due to the combinatorial complexity of potential drugs and dose ranges. As a result, doctors and clinicians often make choices based on a variety of rational criteria such as pharmacokinetic and pharmacodynamic compatibility, lack of side effects, or selecting drugs with divergent MOAs[13–15]. This approach is hindered by an incomplete understanding of MOAs and/or the underlying pathophysiology of the disorder. An empirical alternative to rational polytherapy would involve testing all potential drug combinations in animal models and moving the most promising ones into clinical trials. Such a strategy is impractical for evaluating more than a handful of duotherapy options in standard rodent behavioral models, which are both expensive and low-throughput. High-throughput behavior-based assays in zebrafish have been used in recent years for neuroactive drug screens[16–18] and offer a potential solution for large-scale combinatorial screening. However, behavioral screens are highly reductive, typically employing simple locomotor assays as surrogates for more complex neurological states. Additionally, even behavioral assays with very high throughput remain unsuitable for combinatorial screening of large numbers of compounds. Approximately 1500 drugs have received approval from the U.S. Food and Drug Administration[19]. Empirically testing all possible duotherapy options even in a single disease model would involve $1.1 \times 10^6$ combinations (i.e. ~$10^7$ animal tests). Expanding polytherapy screens to include additional disease models, multiple drug concentrations, larger chemical libraries, or to explore three- or four-way combinations further increases the difficulty of such screens.

Here, we describe an alternative polytherapy screening strategy that addresses these challenges using brain-wide functional connectivity patterns. We apply our approach to a zebrafish model of Dravet syndrome, an intractable genetic epilepsy in humans. Rather than directly testing all possible drug combinations, we first identify how only the individual drugs alter the functional connectivity between brain areas. Broadly speaking, two areas are considered functionally connected if the time series of their neuronal activity (as measured by a genetically encoded calcium indicator (GCaMP)) is highly correlated. Such functional connectivity has been extensively investigated over the past two decades in humans and animal models, typically using functional magnetic resonance imaging (fMRI). Changes in functional connectivity have been described in numerous neurological and psychiatric disorders, including epilepsy, Alzheimer's disease, autism, schizophrenia, major depressive disorder, and post-traumatic stress disorder[20–27]. Additionally, connectivity is altered in short- and long-term responses to neuroactive substances[28,29], meaning that it offers a direct multi-parametric readout of how brain activity is affected by pharmacological interventions. In humans, functional connectivity studies confirm that even focal epilepsies produce widespread changes in connectivity patterns[20], and we report here for the first time the presence of such nontrivial network dysfunctions in epileptic zebrafish brains. Next, using a novel clustering algorithm, we classify drugs based on their functional connectivity fingerprints to identify hits that correct complimentary facets of the abnormal brain-wide network in mutant zebrafish. This allows us to identify polytherapy combinations that are likely to produce synergistic effects based on their ability to target distinct aspects of the underlying network dysfunction, rather than simply combining drugs with different suspected MOAs. Having thus reduced the tremendously large parameter space to a manageable number of options, all top polytherapy combinations can be easily evaluated over a range of doses and tested in follow-up assays for efficacy and side effects. Our approach offers a powerful new classification means for drugs based on in vivo brain activity patterns, which is in many respects complementary to traditional MOA-based approaches.

Although our approach is likely applicable to different animal models, we tested it in zebrafish larvae by taking advantage of the recent advances in light-sheet microscopy for fast cellular-resolution imaging of neuronal activity[30–32]. In order to rapidly assess the effects of compounds on in vivo brain activity with high spatiotemporal resolution, we implemented (1) a high speed light-sheet microscope that can image the brain at single-cell resolution in 50 ms (~15 % of all neurons in each zebrafish larva's brain is imaged), (2) a high-throughput fluidic platform capable of handling and imaging large numbers of larvae under multiple treatment conditions, and (3) algorithms that automatically register the resultant data to a 3D anatomical zebrafish brain atlas for brain-wide connectivity fingerprint analysis. Among the hits that we identified using our approach, the best duotherapy combinations achieve significantly greater seizure reduction than any monotherapy alone. Even at their highest tolerated doses, we find that monotherapies are unable to match the efficacy achieved by our best polytherapy regimens at substantially lower concentrations with minimal side effects. Our results demonstrate the power of network functional connectivity analysis for the discovery of neuroactive drugs, and in particular for polydrug screening.

## Results

**High-throughput mapping of brain activity patterns**. To enable large-scale drug screens based on neural activity in zebrafish expressing genetically encoded calcium indicators, we designed and built a high-speed light-sheet microscopy platform paired with peripheral fluidics for rapidly processing large numbers of larvae under multiple treatment conditions (Fig. 1). The illumination and detection arms of our platform expand on previous light-sheet microscope designs[30] and are detailed in the Methods section. In order to immobilize and precisely position non-anesthetized non-paralyzed zebrafish larvae for analysis, we devised a process through, which larvae can be rapidly embedded in dual-layer agarose cylinders (Fig. 1; see Methods section for

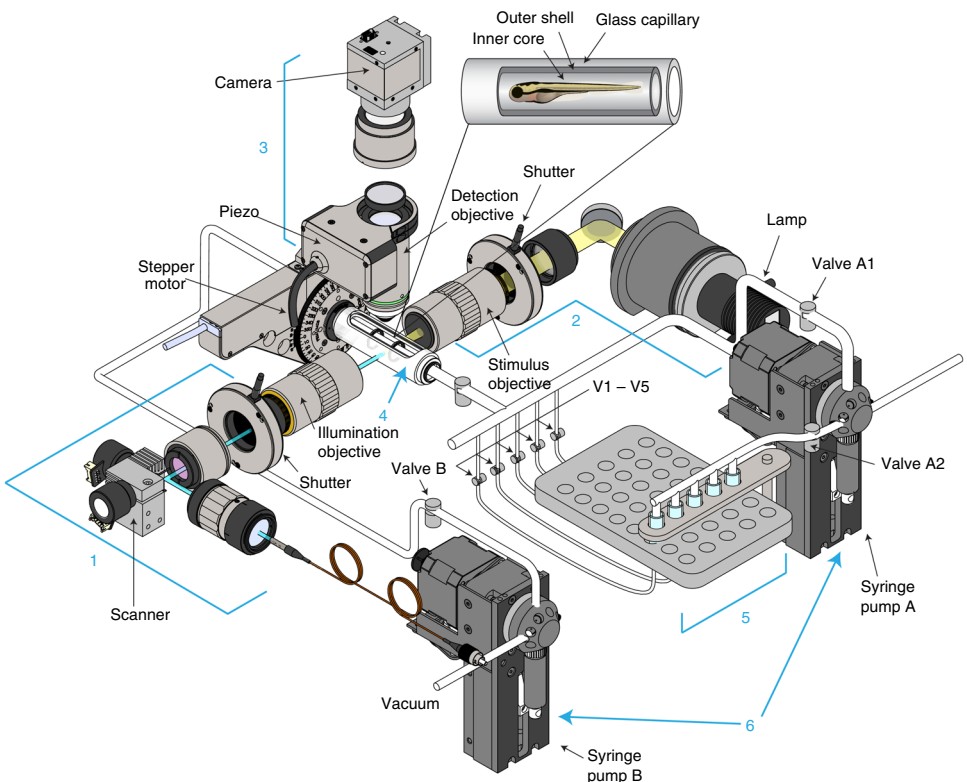

**Fig. 1** High-throughput platform for functional brain connectivity-based drug screening. The platform consists of multiple incubation reservoirs in which zebrafish larvae (captured in a dual-layer polymer pellet) can be treated with compounds of interest and rapidly transferred back and forth and positioned in the imaging chamber of a high-speed light-sheet microscope. The platform has six parts: (1) an illumination arm consisting of a 488 nm fiber laser, a 2D galvo scanner, a tube lens, a shutter, and a low NA ×4 objective. (2) A light-stimulus arm, consisting of a halogen lamp, a filter, a tube lens, a fast shutter to deliver brief light pulses, and a ×4 objective. (3) A detection arm, consisting of a ×16 water immersion objective, a piezo scanner, a 525 nm fluorescent filter, a tube lens, and a high speed sCMOS camera. (4) An imaging chamber, consisting of an ultra-thin wall glass capillary running through a custom-made watertight glass-sided aluminum chamber. The glass capillary is connected to a stepper motor for rotational alignment of the larva. (5) Multiple reservoirs for incubating agarose-embedded zebrafish larvae in drugs. When an incubation chamber is connected to the imaging chamber (via opening a series of pinch valves V1-V5), a syringe pump loads larvae that have been pre-embedded in a dual-layer polymer pellet into the imaging chamber one at a time. (6) Two syringe pumps used for transferring larvae between incubation reservoirs and the imaging chamber and for cleaning the imaging chamber between each imaging cycle

details)[33]. Embedded larvae remain healthy and develop normally for up to 12 h, allowing long-term assessment of the effects of drug-induced changes on brain activity (Supplementary Fig. 1). The outer diameter of the agar cylinders is size-matched to the imaging chamber capillary and all peripheral fluidic tubing. When not being imaged, agar-embedded larvae are housed in incubation reservoirs connected to the light-sheet imaging chamber through a series of fluidic tubes and valves. Initially, incubation reservoirs are filled with E3 embryo medium. All larvae from a reservoir are loaded in turn into the light-sheet imaging chamber using computerized syringe pumps as explained in the Methods section and illustrated in Supplementary Fig. 2. Each larva is imaged to acquire pre-drug-exposure neural activity data and the image is registered to an atlas (see Methods section). Once all larvae from an incubation reservoir have been imaged, they are returned to the reservoir and the drug of interest is added. After a 4+ h incubation time, the entire loading and imaging process is repeated to assess post-exposure neural activity. Once the second imaging session is complete, larvae can either be returned to the incubation reservoir or, as in the following experiments, collected for genotyping. By utilizing multiple incubation reservoirs in a staggered manner, it is possible to achieve continuous imaging of samples. The number of larvae that can be housed per reservoir depends in part on the desired experimental parameters. In the following experiments, we use

five larvae per reservoir and an imaging time of 18 min per larva, meaning that all larvae in a given reservoir are analyzed between 4.0 and 5.5 h post-exposure with ~90 min separating the first and last larva imaged. For experiments requiring greater temporal precision, fewer larvae can be distributed between a larger number of incubation reservoirs.

**Brain-wide imaging reveals abnormal functional connectivity.** In humans, mutations in the voltage-gated sodium ion channel alpha subunit 1 (SCN1A; OMIM *182389) are associated with a spectrum of childhood neurological dysfunctions including Dravet syndrome[34,35]. In zebrafish, the scn1lab gene encodes a voltage-gated sodium ion channel orthologous to mammalian SCN1A and related SCN family members. The scn1lab loss-of-function mutation s552 (previously referred to as double indemnity or didy) causes spontaneous seizures in homozygous mutant zebrafish larvae beginning at ~4 dpf. Mutants respond to many of the same anti-epileptic drugs (AEDs) used to treat Dravet syndrome in humans[16]. In addition to spontaneous seizures, we find that scn1lab mutants experience photosensitive seizures that can be triggered by light pulses, as we recently demonstrated[33]. In order to deliver precisely-timed light stimuli, we use a computer-controlled LED illuminator programmed to administer two brief (500 ms) light pulses separated by a 1 s

interval. When homozygous scn1lab mutant larvae are exposed to light stimuli, forebrain local field potential (LFP) recordings display multiple high-amplitude peaks commencing shortly after the onset of each stimulus and persisting for several seconds (Supplementary Fig. 3a). Similar spontaneous seizure-like bursts are observed periodically in mutant larvae in the absence of light stimuli (Supplementary Fig. 3b), as described previously in the literature[16]. In contrast, sibling controls (i.e. age-matched larvae from the same clutch that are either wild-type or heterozygous for the s552 allele) have a markedly different response, characterized by single lower-amplitude peaks (Supplementary Fig. 3a) that become progressively diminished with each subsequent presentation of the light stimulus (Supplementary Fig. 3c). Additionally, mutants exhibit short rapid bursts of seizure-like locomotor activity that persist for ~5 s after each stimulus, whereas sibling controls show almost no perceptible increase in locomotor activity (Supplementary Videos 1 and 2). To quantify light-triggered swimming behavior, we record larvae under infrared illumination using an automated tracking platform and calculate the mean swimming velocity during 5-s intervals commencing with the onset of each light stimulus. Light-triggered locomotor activity is significantly higher in mutants than in siblings (Supplementary Fig. 4). Importantly, light-triggered seizure-like bursts are observed in LFP recordings from mutant larvae that have been paralyzed with pancuronium bromide, confirming they are not attributable to motion artifacts due to increased locomotor activity in mutants (Supplementary Fig. 5). Photosensitivity appears to be a general feature of scn1lab loss-of-function mutations, as we observe similar abnormal light-triggered phenotypes in both the original scn1lab[s552] line (which contains a p.Met1208Arg missense mutation in ion channel domain III) and in a second scn1lab line (sa16474) that was generated as part of the Zebrafish Mutation Project (Supplementary Fig. 4). This second line contains a C to A mutation at position 1386 of the scn1lab open reading frame, resulting in a premature stop codon at the position between ion channel domains I and II and presumably resulting in a nonfunctional protein[33]. This photosensitive phenotype is consistent with Dravet syndrome in humans, where photosensitive seizures have been reported in 30–40% of patients and are often associated with more severe outcomes[36–38].

We took advantage of photosensitivity, which allows us to control the precise timing of seizures, to image brain activity patterns before, during, and after seizures using light-sheet microscopy. This was done by first crossing the scn1lab[s552] mutation onto a transgenic line expressing the genetically encoded calcium indicator GCaMP5G under the control of the pan-neuronal elavl3 promoter [Tg(elavl3: GCaMP5G)][39]. At 5 dpf, larvae are embedded in dual-layer agarose cylinders and loaded into our high-throughput light-sheet platform for brain-wide activity imaging. In order to analyze neural activity under various states, imaging is carried out over the course of 18 min using the parameters outlined in Fig. 2a. For the first 10 min, larvae are imaged in the absence of any seizure-inducing white light stimuli (pre-stimulus state) to assess resting state activity. Seizures are then triggered in mutant larvae using the stimulus parameters described previously (two 500 ms pulses of light separated by a 1 s interval), which are presented every 2 min over the course of the remaining 8 min For subsequent analysis, this portion of the imaging session is separated into an early post-stimulus state (0–60 s after the presentation of each light stimulus) and a late post-stimulus state (60–120 s after presentation), as indicated in Fig. 2a. Immediately following light stimuli, we see a significant increase in simultaneous GCaMP fluorescence activity in mutant larvae relative to wild-type sibling controls (Fig. 2b, Supplementary Video 3–6). To confirm that abnormal

light-triggered GCaMP activity in scn1lab mutants correlates with electrophysiological hallmarks of seizures, we recorded forebrain LFPs from agar-embedded larvae subjected to the same light stimulus parameters (Fig. 2a), allowing us to compare timestamp-synchronized LFP and GCaMP recordings in independent larvae (Fig. 2b).

Although light-triggered increase in GCaMP fluorescence offers a reliable metric for detecting seizures, it remains a highly reductive readout that fails to take into account alterations to functional networks during non-seizing intervals and in response to drug treatment. Functional connectivity, which is defined as the temporal correlation in activity between spatially separated brain areas, offers a powerful multi-parametric tool for comprehensively analyzing brain-wide functional imaging data-sets. Importantly, functional connectivity maps are known to be altered under various behavioral states[27], making them particularly useful for evaluating how drugs modulate multiple aspects of brain activity and for benchmarking outcomes relative to wild-type connectivity maps. Although functional connectivity is typically studied using fMRI, which uses blood oxygenation level-dependent (BOLD) contrast imaging as an indirect readout of neural activity[40], we instead employ GCaMP5 to directly monitor the calcium signals that reflect neuronal spiking with much higher temporal and spatial resolution. Calcium signals have previously been used to map long-range functional connections between cortical areas in GCaMP3 transgenic mice[41], and a recent study confirms that both calcium indicators and hemodynamic signals yield similar spatial maps of connectivity across the entire brain in mice[42].

In order to use functional connectivity patterns for drug screening, we first use k-means clustering[43] to identify individual neurons (supervoxels) in the image and then we determine the correlation of activity between all pairs of neurons in all areas in our brain atlas (see Methods for details). Finally, we calculate the mean absolute correlation coefficient in order to generate a single metric for each pair of areas during each state. Using this approach, we find that there are clear differences in functional connectivity between scn1lab mutants and sibling controls during all three behavioral states: pre-stimulus, early post-stimulus, and late post-stimulus (Supplementary Fig. 6).

**Identifying neuromodulators to normalize brain connectivity.** In order to identify the most promising compounds for anti-epileptic screening, we previously used a simple behavioral metric (mean light-triggered locomotor velocity) to rapidly evaluate a diverse chemical library and identify compounds with potential therapeutic activity in the photosensitive scn1lab[s552] line[33]. Our starting library contained 154 compounds including AEDs, neuroactive compounds targeting a wide spectrum of neurotransmitter pathways, and compounds that we identified as potential binders to human SCN1A and SCN8A based on an in silico analysis of protein pocket similarity[44,45]. Prior to the screening, all compounds were first assessed for toxicity at a concentration of 100 μM. Those exhibiting overt toxicity at 4 h post-exposure based on reduced/absent touch response were retested at lower concentrations until a maximum tolerated dose was found. From this collection, we selected 24 compounds for in-depth functional connectivity analysis based on their ability to reduce abnormal light-triggered locomotor activity in mutants by at least 50% (Supplementary Table 1).

We then performed neural activity imaging to evaluate the ability of these compounds to correct abnormal connectivity in mutant larvae. Compound screening is carried out on 5 dpf larvae embedded in dual-layer agarose cylinders using a minimum of

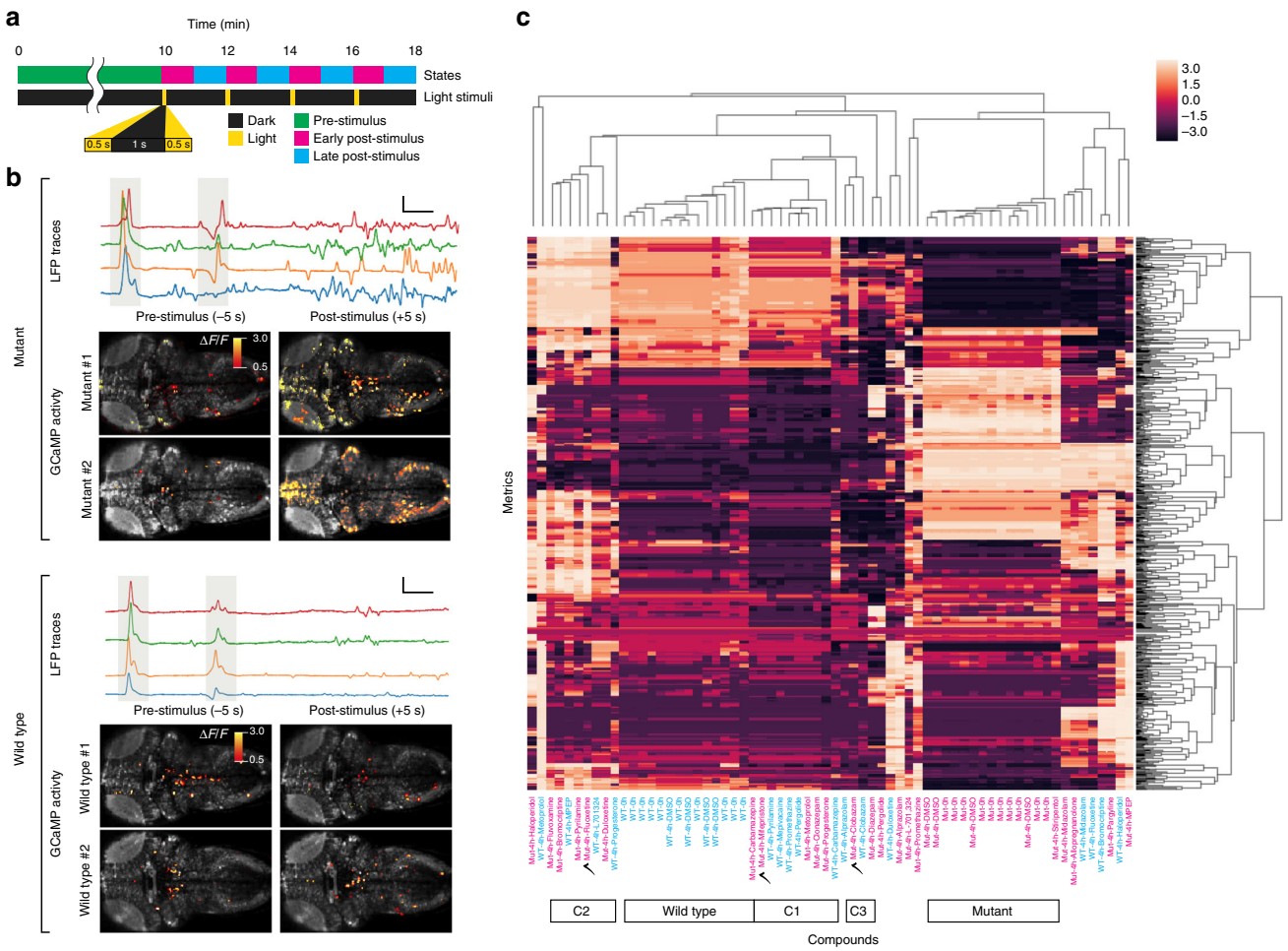

**Fig. 2** Functional brain connectivity fingerprints identify polytherapy candidates. **a** Schematic representation of the light stimulus parameters and associated activity states used for functional connectivity fingerprinting. GCaMP activity is recorded for 10 min in the absence of light stimuli (pre-stimulus state). Seizure-inducing white light stimuli are then applied every two minutes. GCaMP recordings during this period are subdivided into early post-stimulus (0–60 s after the stimulus) and late post-stimulus (60–120 s after the stimulus) states during subsequent analysis. **b** Representative local field potential recordings (LFP traces; upper panels) and single frames from brain-wide GCaMP recordings (GCaMP activity; lower panels) in scn1lab mutants (top) and wild-type controls (bottom) illustrating response to light stimuli at 5 dpf. LFP trances and GCaMP activity are recorded independently. Gray bands on LFP recordings indicate the timing of the two 500 ms light pulses. Scale bars show 0.5 s (x-axis) and 1 mV (y-axis). GCaMP images show activity from a single z-plane 5 s before the first pulse (pre-stimulus) and again 5 s after the second pulse (post-stimulus). Supervoxels with significant changes in fluorescence activity (ΔF/F) are shown in color. **c** For each compound, a 165-metric functional connectivity fingerprint is generated based on correlation coefficients between active supervoxels in all 55 pairs of brain areas during each of the three activity states. Each square represents the deviation of the correlation for the compound in question ($X$) from the mean correlation for all compounds ($X_{mean}$) in the brain area pair being analyzed, normalized by standard deviation [$(X-X_{mean})/X_{STD}$]. The analysis is carried out in both mutants (Mut, magenta text) and sibling controls (WT, cyan text) prior to compound addition (0 h) and again at 4 h post-exposure to the compounds in 1% DMSO (4h). Fingerprints are analyzed by hierarchical clustering to identify compounds that modulate functional connectivity networks in similar ways. We identify the three most distinct clusters (C1, C2, and C3) that fall closest to the wild-type cluster. We then select the compound in each activity cluster that most effectively restores functional connectivity in scn1lab mutants based on Euclidean distance from the wild-type cluster (indicated by checkmarks). Source data are provided as a Source Data file

five homozygous mutants and five sibling controls per compound. Prior to compound exposure, light-sheet imaging is performed on all larvae to establish baseline functional connectivity data for each test group. During pre-exposure imaging, homozygous mutants can be reliably distinguished from sibling controls based on the intensity and duration of GCaMP activity. Immediately after the acquisition of the baseline recording, embedded larvae are transferred from the light-sheet imaging chamber into drug incubation chambers containing test compounds at the specified concentrations using our high-throughput platform. Larvae are incubated in test compounds plus 1% dimethyl sulfoxide (DMSO) for 4+ h, at which point they are transferred back into the imaging chamber and a post-exposure

recording is performed. During both baseline and post-exposure recordings, larvae are subjected to the light-stimulus parameters described previously (Fig. 2a). Following post-exposure imaging, larvae are collected and processed for PCR genotyping, as it is not always possible to distinguish sibling controls from mutants that have been treated with highly effective compounds based on visual inspection of the GCaMP response.

A unique connectivity fingerprint is then generated for each compound that incorporates normalized inter-area correlations from all three states: pre-stimulus, early post-stimulus, and late post-stimulus. The resultant fingerprint comprises 165 metrics (i.e. 55 pairs of brain areas; three states), with each metric representing the average of 5+ larvae. We then use hierarchical

clustering to compare functional connectivity networks in drug-treated larvae. When cluster analysis is carried out on compound-treated wild-type controls, we observe a tendency for drugs with related mechanisms of action to cluster together (Supplementary Fig. 7). For example, both selective serotonin reuptake inhibitors (SSRIs; fluoxetine and fluvoxamine) in our library localize to Cluster 1, four out of five benzodiazepines (alprazolam, clobazam, clonazepam, and diazepam) along with two another GABA agonists (allopregnanolone and stiripentol) localize to Cluster 3, and both glutamate receptor antagonists (L-701,324 and MPEP) localize to Cluster 4. However, we note that there are also a number of other drugs that do not cluster according to their MOA, perhaps indicating an incomplete mechanistic understanding of their in vivo effects.

Adding scn1lab mutants to the analysis allows us to compare functional connectivity between wild-type and mutant networks. As expected, untreated siblings and untreated mutants localize to highly divergent clusters on the resultant dendrogram, confirming that the activity profiles of these two groups are strikingly different (Fig. 2c, Supplementary Fig. 8 and Supplementary Fig. 6). Importantly, multiple independently-imaged groups of untreated siblings (labeled WT-0h on Fig. 2c) cluster with one another, as do multiple independently-imaged groups of untreated mutants (Mut-0h on Fig. 2c), verifying that functional connectivity patterns are relatively uniform for a given experimental condition and therefore provide a robust basis for assessing neuromodulators. Ranking connectivity fingerprints based on Euclidian distance from untreated sibling controls reveals that all 24 compounds produce profiles in mutants that are at least somewhat closer to siblings than to untreated mutants, consistent with the fact that these compounds were selected based on efficacy in a preliminary behavioral assay (Supplementary Table 2).

To maximize the chances of achieving synergistic efficacy in polytherapy, each drug employed should correct distinct aspects of the underlying disorder. Combining multiple drugs that target the same pathological features is likely to increase side effects while producing little additional therapeutic benefit. Choosing drugs with different known MOAs is a common strategy to achieve this result, albeit an imprecise one that depends on an accurate understanding of all biological targets of the drugs. Connectivity fingerprints provide a detailed multi-parametric readout of a compound's actual in vivo effects that can be easily benchmarked against wild-type controls, making them an attractive alternative to MOAs for polytherapy selection. In order to identify polytherapy candidates from our screen that are likely to normalize complementary facets of the underlying pathology, we devised a selection strategy based on multiple rounds of hierarchical cluster identification and elimination. Starting with our initial connectivity fingerprint dendrogram, we first identify the closest cluster to the cluster of untreated sibling control (designated C1 on Fig. 2c). We then select the compound on this cluster that is most effective at normalizing connectivity in scn1lab mutants based on distance from the sibling cluster (mifepristone in this case). In the next step, we eliminate all compounds that localize to this cluster from further analysis. Regardless of their MOA, these compounds modulate pathological brain activity in a highly similar manner and therefore represent poor candidates for pairing with the top hit in a polytherapy regimen. We then perform a new round of clustering on the remaining compounds and repeat the previously described selection process to identify a new hit. The second and third rounds of compound selection and cluster elimination identified fluoxetine and clobazam as top hits, respectively (Supplementary Fig. 9). As expected based on their ability to differentially modulate functional connectivity networks, each of these three compounds belongs to different drug classes (Supplementary Table 1).

**Algorithmic combinatorial screening for superior polytherapy.** Having reduced the polytherapy parameter space to a manageable size, we evaluated all duotherapy combinations for both efficacy and behavioral side effects over a range of doses. This was done using LFP recordings to quantify spontaneous seizures and automated behavioral phenotyping to detect compound-induced alterations in multiple locomotor parameters. We tested all possible pairwise combinations of our top three hits (clobazam, fluoxetine, and mifepristone) in both assays using a semi-log range of doses: 10, 30, and 100% of the concentration used for light-sheet screening. For LFP analysis, we assessed the ability of each individual compound and duotherapy combination to reduce spontaneous seizures in mutants using an automated seizure detection algorithm we previously developed based on methods for analyzing EEG signals[33,46]. Baseline seizure frequency was first determined for each larva during a 45-min pre-exposure LFP recording. Seizure frequency was measured again during a second recording beginning at 4+ hours post-exposure and an efficacy score was calculated by normalizing the post-exposure frequency to the baseline frequency (Supplementary Table 3, Efficacy column).

In order to measure compound-induced side effects and behavioral alterations, we performed an in-depth behavioral assessment for all 36 treatment conditions relative to untreated (1% DMSO-only) scn1lab mutants. Behavioral assessment was carried out in 96-well plates using an automated video tracking system and algorithms we developed previously[33]. Compounds were applied directly to the wells and 30-min video recordings were carried out at ~4 h post-exposure. We then automatically quantified six distinct behavioral features related to swimming velocity and tail motion for each larva: (1) Mean forward swimming velocity, (2) Mean angular swimming velocity, (3) Standard deviation of the angular swimming velocity, (4) Time spent at high swimming velocity, (5) Mean tail bending angle, and (6) Mean change in tail angle. All behavioral features were normalized to baseline levels in untreated mutants (Supplementary Table 3, Behavioral Metrics columns). An overall side effect score was then defined for each treatment condition by determining how much each behavioral feature diverged from its baseline value in untreated controls and calculating the root mean square of all six divergences (Supplementary Table 3, Divergence from Baseline and Side Effect columns). The side effect score therefore measures the overall magnitude of behavioral changes triggered by an individual compound or duotherapy combination.

LFP and behavioral assessments confirm that polytherapy combinations identified by connectivity fingerprint analysis are able to achieve significantly greater efficacy than any monotherapy alone (Fig. 3, Supplementary Table 3). Our data also clearly shows that combining the top hits from a behavioral screen alone, even after selecting those with different known MOAs, performs far worse than our connectivity-based polytherapy (Supplementary Fig. 12). Importantly, a number of duotherapy regimens show significantly greater therapeutic efficacy at lower total concentrations while maintaining side effect profiles comparable to monotherapies. Fluoxetine-mifepristone duotherapy appears to be particularly effective, with multiple lower-dosage combinations displaying substantially greater efficacy than even the highest doses of either drug on its own (Fig. 3, Supplementary Video 7).

To test whether polytherapy indeed restores normal brain connectivity better than monotherapy, we carried out light-sheet imaging on mutants treated with fluoxetine or mifepristone alone at 100% concentration and compared the results with one of our optimal duotherapy regimens (30% fluoxetine, 30% mifepristone). Larvae were assessed under the light-stimulus parameters

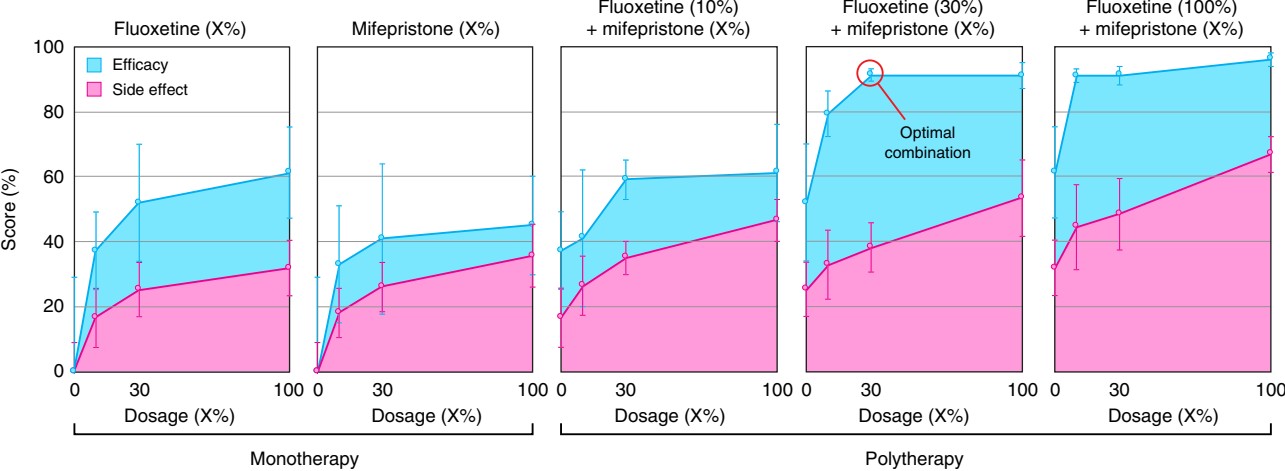

**Fig. 3** Network-engineered polytherapy achieves superior therapeutic outcomes. Semi-log dose-response data showing efficacy scores (cyan) and side effect scores (magenta) for fluoxetine and mifepristone alone (monotherapy) and combinatorially over all investigated doses (polytherapy) at ~4 h post-exposure. For each compound, the highest dosage (100%) corresponds to the concentration used in the initial light-sheet-based screen (see Supplementary Table 1). Efficacy is calculated from local field potential recordings using an automated seizure detection algorithm to count spontaneous seizures over a 45-min interval. The side effect score is based on six distinct behavioral metrics quantified over a 30-min interval using an automated video tracking platform. Both scores are determined under resting state conditions (i.e. in the absence of light stimuli) and calculated as described in Methods. Scores have been normalized to untreated age-matched larvae ($n = 5+$ larvae per condition for efficacy; $n = 10$ larvae per condition for side effect). Several polytherapy combinations show substantially greater therapeutic efficacy while maintaining behavioral profiles that are in the same range as monotherapy regimens. The combination of fluoxetine and mifepristone, both at the 30% dosage (optimal combination), is significantly more effective than either mifepristone alone ($p = 0.000106$; unpaired two-tailed Student's $t$-test) or fluoxetine alone ($p = 0.000105$; unpaired two-tailed Student's $t$-test) at the same dosages. The error bars represent standard deviation. Source data are provided as a Source Data file

described previously (Fig. 2a) and connectivity fingerprints were generated for each drug regimen. To evaluate the ability of neuromodulators to restore resting state functional connectivity patterns, we first identified all significantly correlated pairs of brain areas in wild-type larvae (24 out of 55 total pairs; Fig. 4a). For each pair of areas, we then calculated the resting state divergence in connectivity (i.e. distance from wild-types in the resting state) for each drug regimen (Fig. 4b–e; Supplementary Fig. 10a). In addition, we calculated the divergence in connectivity in the other two behavioral states (i.e. early post stimulus and late post stimulus) for each drug regimen (Supplementary Fig. 10b–c). Functional connectivity differed substantially from wild-type larvae in nearly all brain area pairs in untreated (DMSO) scn1lab mutants (Fig. 4b). Out of the 24 connections evaluated, 11 showed decreased connectivity and eight showed increased connectivity relative to wild-types, while five connections were not altered significantly. Both fluoxetine and mifepristone alone reduced abnormal connectivity patterns to varying degrees, however their efficacy was dramatically increased when used in combination (Fig. 4e; Supplementary Fig. 10).

## Discussion

There is a growing consensus that most neurological and psychiatric disorders are associated with large-scale dysfunction of brain connectivity[47]. In epilepsy, pathologic networks have been used to explain seizure generation and spread, cognitive impairment, and therapeutic response[20]. fMRI studies, sometimes paired with simultaneous EEG, provide clear evidence for functional connectivity network abnormalities in epileptic brains[48]. The epileptogenic network in temporal lobe epilepsy (TLE), the most common type of epilepsy, has been extensively characterized. Many fMRI studies report decreased connectivity in patients with TLE relative to healthy controls, although there are also indications of increased connectivity between some areas[48]. The literature remains divided over the precise nature of these changes, with some studies reporting decreased connectivity

within the epileptogenic region and increased connectivity to other areas of the brain (perhaps reflecting compensatory mechanisms)[49,50], while others report the opposite pattern[51]. In addition to altered connectivity, there is evidence that anti-correlated activity in areas outside the epileptogenic region may represent a protective adaptation that helps to limit seizure spread[52]. Taken as a whole, these findings demonstrate that epilepsy disrupts neural connectivity patterns throughout the brain, causing both compensatory adaptations and deleterious changes in the functional network. Functional connectivity therefore represents a direct and exceptionally rich method for assessing the underlying pathologies of epilepsy and other neurological disorders. As such, new approaches for analyzing network connectivity hold great promise for both clinical diagnosis and drug discovery.

In the present study, we describe a high-throughput platform for functional connectivity analysis and report for the first time that neuronal network dysfunctions similar to those observed in human patients underlie epilepsy in a vertebrate model (zebrafish) that is amenable to large-scale screening. Our approach to analyzing functional brain connectivity in GCaPM5-expressing scn1lab mutant zebrafish larvae builds on strategies that have been successfully employed in humans to compare connectivity between healthy controls and epileptic patients using fMRI[53,54]. As in human studies, we begin by segmenting the registered GCaMP recordings into areas based on a 3D anatomical brain atlas. However, rather than averaging all voxels within each area prior to correlation analysis, the superior spatial resolution afforded by the GCaMP signal allows us to directly analyze all spatially distinct supervoxels (presumably corresponding to individual neurons) within each area. Additionally, the ability to optically trigger seizures on demand in scn1lab mutant zebrafish enables us to assess connectivity during distinct, pathologically relevant brain activity states. The resultant connectivity fingerprints offer a highly accurate multi-state, multi-parametric tool that readily distinguishes between wild-type and mutant larvae, as

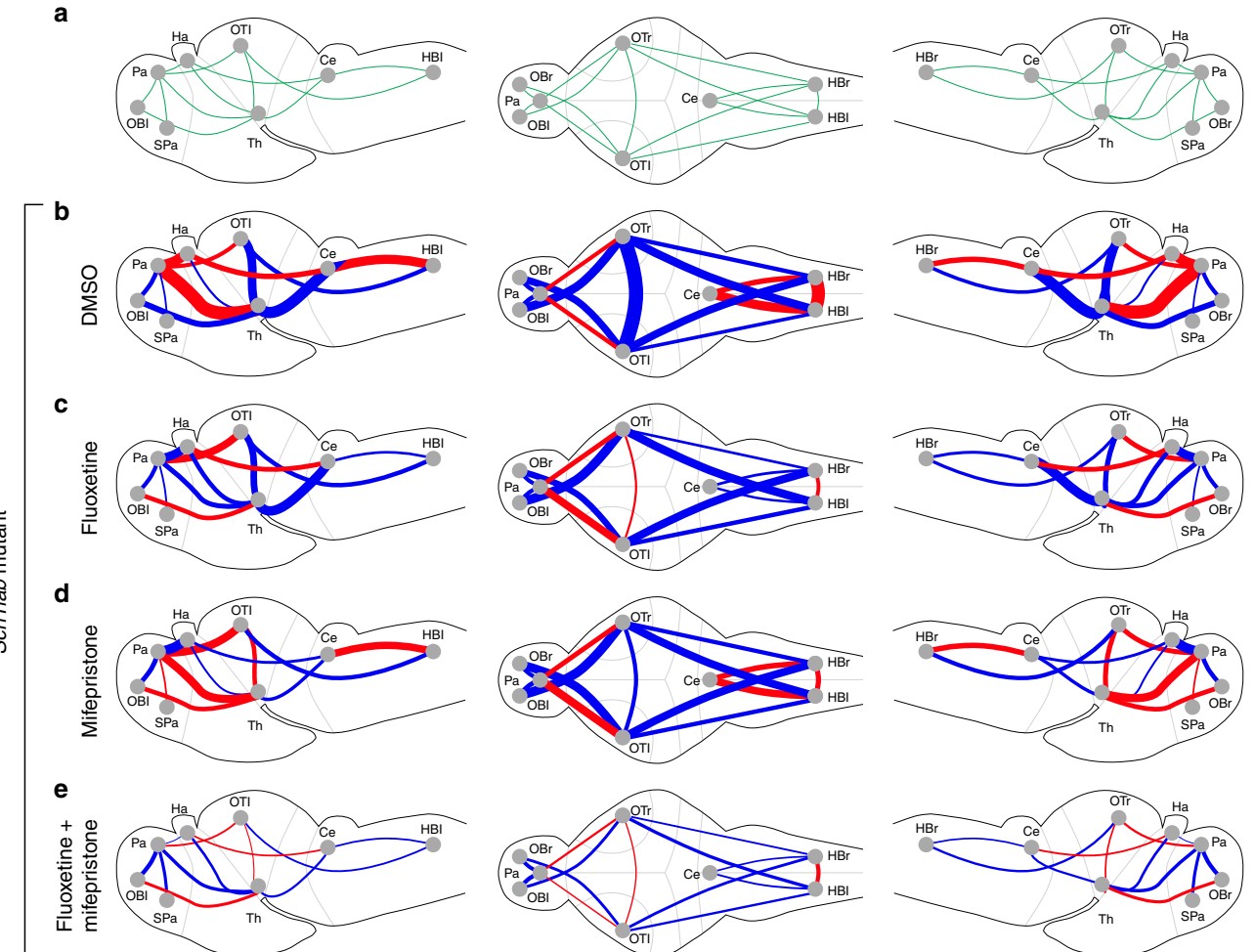

**Fig. 4** Network-engineered polytherapy restores resting state brain connectivity in mutants. **a** Diagram depicting brain areas with statistically significant absolute correlation values (unpaired two-tailed Student's t-test; $p < 0.05$ considered significant; normality of datasets was tested using the Jarque–Bera normality test) in wild-type larvae at 5 dpf ($n = 10$). Only these connections are shown in **b**–**e**. **b**–**e** Diagrams illustrating alterations in functional brain connectivity in scn1lab mutant larvae with respect to wild-type sibling controls under the following conditions: **b** untreated mutants (DMSO), **c** mutants treated with fluoxetine alone at the 100% dose, **d** mutants treated with mifepristone alone at the 100% dose, and **e** mutants treated simultaneously with fluoxetine and mifepristone using one of the optimal polytherapy dose regimens (both drugs at 30%). For all diagrams, brains are shown from the left side (left), from the dorsal side (center), and from the right side (right). Connections which increase or decrease with respect to wild-type larvae are indicated with blue or red lines, respectively. The thickness of the lines in b–e indicates the magnitude of deviation from normal wild-type connectivity. Functional connectivity for each pair of brain areas is determined at 4+ h post exposure by calculating the average absolute correlation between active supervoxels within each area during the pre-stimulus states. For each compound, the highest dose (100%) corresponds to the concentration used in the initial light-sheet-based screen (see Supplementary Table 1). Ten larvae were analyzed per condition. Ce, cerebellum; HBl, hindbrain left; HBr, hindbrain right; OBl, olfactory bulb left; OBr, olfactory bulb right; OTl, optical tectum, left; OTr, optical tectum, right; Pa, pallium; SPa, sub-pallium; Ha, Ha, habenula; Th, thalamus (For ease of illustration, the areal labels/borders are not always shown on anatomically correct positions/scales)

confirmed by cluster analysis (Fig. 2c, Supplementary Fig. 6). The classification accuracy of our approach when compared with similar attempts in human patients undoubtedly benefits greatly from the ability to control virtually all experimental conditions: the zebrafish larvae used are medication-naive age-matched siblings and their seizures are driven by a common mutation to a single human disease-associated gene.

Using our high-throughput platform, we go on to demonstrate how brain-wide functional connectivity analysis can be utilized to overcome the massive combinatorial challenge of polytherapy screening. By classifying drugs based on their actual in vivo effects on brain networks, we are able to identify highly effective candidates for combination therapy even in the absence of information on underlying MOAs. Such an empirical approach to polytherapy selection based on connectivity analysis has the potential to benefit patients suffering from a broad range of

disorders. In addition to epilepsy, aberrant functional connectivity networks have been associated with autism[21,22], schizophrenia[23,24], major depressive disorder[25], post-traumatic stress disorder[26], bipolar disorder[55], and attention-deficit hyperactivity disorder[56] among others. Moreover, connectivity can be monitored using a variety of readouts (calcium signals, fMRI, high-density EEG) that are suitable for animal screens and even human studies. Our connectivity-based approach to assessing and classifying neuromodulators can therefore be applicable to a wide range of disease models and may even prove useful for the clinical implementation of personalized medicine in neuropsychiatric disorders.

Although neither of the drugs in our network-engineered polytherapy regimen is classified as an AED clinically, anticonvulsant effects have been reported for fluoxetine (Prozac), which is an antidepressant of the selective serotonin reuptake

inhibitor class. According to a recent report, fluoxetine was effective in markedly reducing seizures in an adult woman with Dravet syndrome who had previously failed to respond to numerous other AEDs[57]. Additionally, fluoxetine has been shown to exhibit anticonvulsant effects when administered either acutely or as a dietary supplement in a variety of rodent seizure models[58–61]. Of particular interest in light of our observations, there are indications that fluoxetine may be able to increase the anticonvulsant activity of several common AEDs, including carbamazepine, phenobarbital, diphenylhydantoin, and valproate, against PTZ-induced seizures[62] and in the maximal electroshock seizure model in mice[61]. The second drug in our polytherapy regimen is mifepristone (RU-486), a synthetic steroid that acts as an antagonist of both the progesterone and glucocorticoid receptors in mammals[63] and in zebrafish[64,65]. Anticonvulsant effects have not previously been associated with mifepristone, however there is ongoing interest in targeting glucocorticoid receptors in patients with epilepsy due to the pro-convulsant actions of stress hormones such as cortisol[66]. Our data in zebrafish indicate that additional investigation of both fluoxetine and mifepristone—or related compounds—is warranted in Dravet syndrome and other epilepsies. More generally, our results point to the need to look beyond monotherapy regimens and to utilize more sophisticated readouts that capture the full complexity of epilepsy and other CNS disorders in future drug screens.

It is also worth mentioning that our functional connectivity analysis alone does not imply that a drug cocktail should always lead to behavioral improvement. Rather, it identifies a cocktail that corrects network functional connectivity: In rare cases, correction of functional connectivity may not imply the correction of all phenotypes. For example, carbamazepine ranks highly in our connectivity analysis because it indeed normalizes particular aspects of functional connectivity, however our additional LFP analysis[33] indicates this particular drug doesn't improve seizure in zebrafish. Thus, our functional connectivity analysis should be complemented with other metrics, such as behavioral and LFP analysis, as we did previously[33].

## Methods

**Fish maintenance**. All procedures on live animals were approved by the Massachusetts Institute of Technology Committee on Animal Care. The scn1lab[s552] line (also known as double indemnity or didy) has been described previously[16] and was crossed onto a line expressing the genetically encoded calcium indicator GCaMP5G[39] under the control of the pan-neuronal elavl3/HuC promoter [Tg (HuC:GCaMP5G); a generous gift of A Schier, Harvard, Cambridge, MA]. Homozygous mutant scn1lab[s552] larvae and age-matched sibling controls expressing the GCaMP5G reporter were obtained by crossing Tg(HuC:GCaMP5[+/+]; scn1lab[+/−]) adults. Adult fish were maintained under standard laboratory conditions and larvae were staged as described[67]. Fertilized eggs were generated by crossing heterozygous adults and raised on a 12 h light/12 h dark cycle at 28 °C in E3 medium (5 mM NaCl, 0.17 mM KCl, 0.33 mM CaCl₂, 0.33 mM MgSO₄, pH 7.2). 0.2 mM 1-Phenyl-2-thiourea (PTU) was added to the embryo medium to inhibit melanogenesis[68] and allow optical imaging of the brain for light-sheet experiments.

For LFP and behavioral experiments, homozygous mutant scn1lab larvae were identified based on the presence of dispersed melanosomes[69]. Because PTU prevents the use of the pigmentation phenotype to identify homozygous mutants, all larvae used in light-sheet imaging experiments were genotyped post-imaging using the Derived Cleaved Amplified Polymorphic Sequences (dCAPS) method to detect the scn1lab[s552] mutation[70]. PCR conditions consisted of 37 cycles of 95 °C for 20 s, 60 °C for 20 s, and 68 °C for 30 s. The following PCR primers were used for the dCAPS assay, resulting in the introduction of an BamHI-sensitive restriction site in the mutant but not the wild-type scn1lab allele.
Forward: TGCTCAGGCTGTGTGATGAGG
Reverse: TCACCAGTGCTCCGCTGCTGAGTAGGATC

**Dual-layer agarose embedding**. To securely immobilize and precisely position non-anesthetized non-paralyzed zebrafish larvae for extended light-sheet imaging sessions, we devised a process through which larvae can be embedded in a cylinder of 1.3% ultra-low gelling temperature agarose (which solidifies at 25 °C; A2576, Sigma) surrounded by shell of 2% low gelling temperature agarose (which solidifies at 55 °C; A0701, Sigma). The ultra-low gelling temperature core allows larvae to be

safely added to the agarose while in a liquid state without being exposed to excessive temperatures. The more rigid low gelling temperature agar shell strengthens and supports the inner core, allowing it to be inserted into the glass capillary and ensuring that the larvae are fully immobilized. Embedding is accomplished by first transferring larvae into a solution of liquid 1.3% ultra-low gelling temperature agarose, which is then poured into a 20 mL syringe. The 20 mL syringe is then inserted into a 60 mL syringe filled with 2 % low gelling temperature agarose. The 20 mL syringe is capped with an 18-gauge stainless steel needle and the 60 mL syringe is capped with a 16-gauge stainless steel needle. We then simultaneously extrude both agarose solutions into a room temperature bath containing E3 medium, allowing the agar to rapidly solidify.

**High-throughput light-sheet microscopy screening platform**. The illumination and detection arms of our platform build on previous designs for laser-scanning light-sheet microscopes[30]. We use a 488 nm 100 mW low-noise collimated diode laser (LLD-0488, Laserglow Technologies, Toronto, Canada) with tunable power supply to generate the excitation laser beam. The light sheet is created by a 2D galvanometer-based scanner (6210H, Cambridge Technology, Cambridge, MA) and a tube lens (ITL200, Nikon, Tokyo, Japan). A ×4/0.1 NA objective lens (Nikon, Tokyo, Japan) is used to illuminate the sample with the light sheet. A second ×4/0.1 NA lens is used to direct the seizure-inducing light pulses, which are generated using a Quartz Tungsten-Halogen lamp (QTH10/M, Thorlabs, Newton, NJ) gated by a stepper-motor driven shutter and microprocessor-based controller (Smart-Shutter and Lambda SC controller, Sutter Instrument Company, Novato, CA). Emitted light is detected with a ×16/0.8 NA water immersion objective (Nikon) mounted on a piezo-driven nanofocusing device (P-725 PIFOC, Physik Instrumente, Auburn, MA) followed by 525/50m emission filter (Chroma Technology, Bellows Falls, VT) and a tube lens (ITL200, Nikon). Images are captured on a sCMOS camera (ORCA-Flash4.0 V2, Hamamatsu Photonics, Hamamatsu, Japan). For functional connectivity analysis, images are captured with 4 × 4 binning at 50 frames per second with a 2 ms exposure time. The structural images used for registration to the anatomical atlas are captured at a higher resolution with 2 × 2 binning and a 20 ms exposure time. The camera is controlled through a custom-made program that uses the Hamamatsu plugin for LabVIEW (National Instruments, Austin, TX).

To facilitate the light-sheet-based screening of compound libraries, we developed peripheral fluidics that allow zebrafish larvae under multiple treatment conditions to be repeatedly imaged with a near-continuous workflow (Supplementary Fig. 1). Our setup consists of an array of incubation chambers, which are fabricated by removing the filters from size exclusion spin columns (#7326227, Bio-Rad, Hercules, CA). Chambers are initially filled with 500 μL of E3 medium and used to hold larvae that have been embedded in dual-layer agarose cylinders as described in the previous section. Incubation reservoirs are held in an array of holes drilled into a 5 mm thick plexiglass holder (McMaster-Carr, Elmhurst, IL, USA). Separate peristaltic tubing lines (1.47 I.D. HelixMark silicon tubing, catalog #60–011–06, VWR, Radnor, PA) gated by pinch valves (B4M, Hach, Loveland, CO) are attached to openings at the top and bottom of each incubation chamber. The upper tubing supplies air at 5 PSI when the incubation chambers are drained. All lines from the bottom openings of the incubation chambers are connected to a master peristaltic tube line (1.98 I.D. HelixMark silicon tubing, catalog #60–011–09, VWR), which in turn is connected to an automated 10 mL syringe pump (Cavro 20738449; Tecan, Männedorf, Switzerland) with E3 medium at one end and a thin-wall fused silica glass capillary (Index of reflection of 1.458, 1 mm I.D., 10-SG-1, Charles Supper Company, Natick, MA) at the other end. This glass capillary serves as the imaging chamber and runs through the center of a 1 × 1 × 1 inch³ custom fabricated aluminum chamber (RM1G, Thorlabs). Mounting holes for the capillary are fitted with a pair of quarter-inch bearing balls (01376748, MSC Industrial Supply, Melville, NY) in order to create a water-tight seal around the capillary and allow the aluminum chamber to be filled with water for the water immersion objective. A stepper motor (Nema 17 High Torque Stepper Motor) is used to rotate the capillary and control the angular position of the larva in the imaging chamber. The end of the capillary opposite that which the larva enters is connected to peristaltic tubing (1.47 I.D. HelixMark silicon tubing, catalog #60–011–06, VWR), which contains a pinch valve to regulate flow (B4M, Hach) and attaches to a three-way syringe pump (Cavro 20738449; Tecan). The three-way pump in turn is attached to (1) the vacuum that collects the excess liquid used to wash the imaging chamber after each imaging cycle and (2) a 10 mL syringe that is used to control the lateral position of larva in the imaging chamber.

To load larvae from one of the incubation reservoirs into the imaging chamber, the glass capillary is first flushed to remove any residual liquid from the previous imaging step. This is done by opening pinch valves A1 and B1 and transferring 1 mL of E3 medium from syringe pump A into syringe pump B. Both pinch valves are then closed and the E3 medium is expelled from syringe pump B into a vacuum flask. To load larvae from the n-th incubation reservoir in the array, we open pinch valves V$_n$, A2, and B1. Syringe pump B then drains liquid from the n-th reservoir until the first larva is in the imaging chamber. When the larva is in the place, pinch valves V$_n$, A2, and B1 are closed. After the imaging session is completed, all three valves are opened and continue draining the reservoir until the next larva enters the imaging chamber. This process is repeated until all larvae contained in the n-th

incubation reservoir have been imaged, at which point pinch valves are opened and the larvae are transferred back into the incubation reservoir using syringe pump B.

After completion of the initial pre-drug-exposure imaging session, 500 μL of the test compound (2x working stock prepared in E3 medium; 2 % DMSO) is added to the incubation reservoir. This brings the total reservoir volume to 1 mL, resulting in the indicated screening concentrations (Supplementary Table 1) and a final DMSO concentration of 1 %. At 4+ h post-exposure, all larvae in the incubation reservoir are once again loaded into the imaging chamber for a second imaging session following the same procedure as before. Once the second imaging session is complete, larvae are collected for PCR genotyping.

**Image processing and cluster analysis.** We created a custom 3D anatomical brain atlas based on high-resolution (1024 × 1024 pixel) structural images obtained from GCaMP5-expressing zebrafish larvae at 5 dpf. Structural images were acquired from 20 slices in steps of 20 μm along the z-axis (i.e. the dorsal–ventral axis of the larva) over the course of a ~4 min recording. Manual segmentation was carried out based on anatomical landmarks with the aid of the Atlas of Early Zebrafish Brain Development[71] and the Z-Brain Atlas[72]. Segmentations were performed using ZBrainViewer[72].

To facilitate registration of GCaMP5 recordings to our 3D reference atlas, we first acquire a high-resolution structural reference image of each brain prior to functional activity imaging. Reference images are generated by calculating the average of the voxel baseline over a 200-s interval at the start of each image acquisition. As with structural images used for the brain atlas, these reference images are acquired at 1024 × 1024 pixels (1100 μm × 1100 μm field of view) from 25 slices in steps of 20 μm along the z-axis. The brain is extracted from the background in each image with the FSL Brain Extraction Tool[73] using a threshold on the intensity level. The start of the stack is selected automatically using automated landmark detection to identify the dorsal surface of the optical tectum. The high-resolution reference image is then downsampled and resliced to 256 × 256 pixels. This volumetric 3D reference image is next aligned to the corresponding plane in the structural images from the atlas by rigid transformation using the automated volume-based Advanced Normalization Tools (ANTs) software package[74] in Matlab (Mathworks, Natick, MA) to obtain a rough alignment. The rigid transformation is then applied to the original high-resolution reference image. In the second step, the high-resolution reference image is registered to the atlas structural image using affine transformation. We then compare the accuracy of our segmentation algorithm to that of manual segmentation (Supplementary Table 5). Our result shows our algorithm is highly accurate for clustering fingerprints (Supplementary Fig. 11). To demonstrate the segmentation is sufficiently accurate, we calculated the clusters after randomly modifying the segmentation such that the new segmentation has only 95% overlap with the original segmentation and then we measured the distance between the original C1, C2 (Supplementary Fig. 11a) and C3 (Supplementary Fig. 11b) clusters from Fig. 2c. Our results show even 10% error in the segmentation does not significantly change the clusters.

After the acquisition of the high-resolution structural images, we perform high-speed functional recordings using reduced resolution 512 × 512 pixel images from 10 slices in steps of 40 μm along the z-axis, resulting in a brain-wide imaging rate of 20 Hz. Based on our depth of field (9.5 μm per slice) and the distance between planes, we cover a total scanning range of ~120 μm along the z-axis (i.e. the dorsal–ventral axis). In order to determine functional connectivity between brain areas, we begin by first identifying all voxels that show time-varying activity during the pre-stimulus state. For each voxel, we define the time-averaged signal as:

$$\sqrt{\frac{\sum(\mathbf{f} - \mathbf{f}_0)^2}{T - 1}} \qquad (1)$$

where $\mathbf{f}$ is the vector of intensity of the voxel time, T is the number of samples over time, and $f_0$ is the average of $\mathbf{f}$ over time. Since a single neuron may be covered by multiple voxels, we combine signals from highly time-correlated adjacent voxels using k-means clustering where the number of voxels is determined by k-fold cross-validation[43], resulting in a list of all spatially distinct supervoxels. We then find the correlation coefficient between active supervoxels for all pairs of areas during each of the three behavioral states. A voxel is considered non-time-varying if its time-averaged signal is ≥1 standard deviation below the mean time-averaged signal of all voxels in the same slice. Since a single neuron may be covered by multiple voxels, we combine signals from highly time-correlated adjacent voxels using k-means clustering where the number of voxels is determined by k-fold cross-validation[43], resulting in a list of spatially distinct supervoxels with co-varying activity. We then identify active supervoxels that show a significant change in GCaMP5 fluorescence (ΔF/F). The threshold level for active supervoxel detection in each slice is determined using an open source calcium imaging processing toolbox for the analysis of neuronal population dynamics[75] by fitting a Gaussian process to each supervoxel resting-state time series. Detected supervoxels have an average diameter of 7.1 ± 3.2 μm (mean ± SD), consistent with the size of individual cell bodies. On average, we detect 14,950 ± 583 (mean ± SD) total supervoxels per brain (n = 50 GCaMP recordings from independent larvae) of which ~10 % are designated as active supervoxels. Zebrafish larvae are estimated to have on the order of $10^5$ neurons[76]. Assuming that each supervoxel corresponds to an individual neuron, we are able to detect ~15% of all neurons, consistent with the

fraction of brain that is covered based on our imaging parameters (~9.5 μm thick slices acquired in steps of 40 μm).

Functional signals are extracted from the volumetric data based on the relative change in GCaMP5 fluorescence (ΔF/F). The baseline for each supervoxel (F) is measured by averaging intensity over a 20-s sliding window. We next divide each functional recording into the following three activity states: (1) Pre-stimulus (an initial 10-min resting state recording in the absence of light stimuli), (2) Early post-stimulus (10–60 s after presentation of seizure-inducing light stimuli; the first 10 s after stimuli is removed from analysis to increase the stationarity of signal), and (3) Late post-stimulus (60–120 s after presentation). We then determine the mean absolute correlation coefficient, for each pair of brain areas (e.g. area A and area B) during each of the three activity states as follows. We calculate the mean absolute Pearson's correlation[30] over all the pairs of voxels from area A and B (A vs B correlation).

Here, we demonstrate that the correlation analysis of active supervoxel pairs outperforms a simpler global pooling. First, we calculated the correlation coefficients of global pooling as suggested and then used the results for compound clustering. To compare the quality of clusters, we measured two metrics: the distance between wild type cluster and mutant cluster (BCD, the larger the better), and the distance within the wild type cluster (WCD, the smaller the better). Our results show that both metrics deteriorate when we use global pooling instead of voxel correlation: BCD decreases by ~56% (from 1.62 to 0.91) and WCD increases by ~110% (0.38–0.80). Furthermore, we demonstrate that BCD/WCD ratio decreases if larger voxels were used (Supplementary Fig. 13).

In order to verify that we are analyzing sufficient numbers of neurons per brain area, we tested the stability of functional connectivity metrics. First, we calculate the correlation coefficient under resting conditions (i.e. the 10 min prior to the presentation of seizure-inducing light stimuli) between a pair of brain areas over all active supervoxels. Then, we randomly group the supervoxels in each brain area into two subgroups and recalculate the inter-area correlation coefficients for each subgroup and calculate the difference in correlation. This process is repeated 50 times for each pair of brain areas. We then test if there is a significant difference between the correlation coefficient group differences. The median of the p-values for all 55 brain area pairs in 10 scn1lab mutants indicate no significant differences in correlation coefficient across different subgroups of supervoxels (p < 0.05; n = 10).

In order to verify that our time series connectivity data are stationary, we analyzed GCaMP recordings from 10 scn1lab mutants and 10 wild-types during the pre-stimulus resting state (i.e. the 10 min prior to the presentation of seizure-inducing light stimuli) using the Priestley-Subba Rao test[77]. The analysis is performed using an open source stationarity R package (https://rdrr.io/github/gnardin/stationarity/src/R/priestley.subba.rao.test.R). Statistical tests for non-stationarity require defining a test statistic and an appropriate framework for generating null data. Null hypothesis testing is then performed by comparing test statistics from real data against those from null data. For each brain area pair, we use multiple 5-min blocks assembled randomly from 30-s windows to assess the second order stationary of the correlation coefficient. We test the weak-sense stationarity hypothesis of correlation (i.e. that its first and second order ensemble statistics are constant in time)[78] by determining the variance (κ) of the sliding window correlation (**SWC**):

$$\kappa = \frac{1}{T - 1} \sum_{t=1}^{T} [\mathbf{SWC}(t) - \mu]^2 \qquad (2)$$

where the correlation between two areas is calculated over 30 s windows, **SWC**(t) is the **SWC** at time t, and μ is the mean of the **SWC** time series. Larger κ values relative to the null distribution suggest non-stationarity (i.e. dynamic connectivity). We generate null data by fitting a second order autoregressive randomization (ARR) process to the SWC time series. Data from the original time series are then tested against the ARR null data. In this test, the null hypothesis corresponds to dynamic connectivity. Therefore, If the two datasets are not significantly different (p < 0.05) the null hypothesis is rejected and the original data are assumed to be stationary. For all 20 larvae analyzed, the median p-value of the second order stationarity test for the 55 area pairs was <0.05. On average, time series connectivity data from ~51 of 55 (93%) brain area pairs per larva were found to be stationary (p < 0.05; unpaired Student's t-test) (Supplementary Table 4).

Hierarchical cluster analysis of functional connectivity fingerprints is performed using Scikit-learn (http://scikit-learn.org/stable).

**Behavioral analysis.** Locomotor activity is recorded using a custom-built video tracking system consisting of a monochrome CCD camera (Prosilica GX1050; Allied Vision, Exton, PA) fitted with a close-focusing macro video lens (Zoom 7000 lens system, Navitar, Rochester, NY), a near IR longpass filter (LP800–52, MidOpt, Palatine, IL), and an IR, white light LED illuminator (BX 06 06 WHI/IR, Advance illumination, Rochester, VT). The system is surrounded by a custom-made light-tight optical table enclosure and mounted on an optical breadboard base. The IR/LED illumination is controlled by an Arduino Mega 2560 microcontroller board (digital output range from 0.0 to 5.0 volts; Adafruit Industries, New York, NY).

At 5 dpf, homozygous mutant larvae are separated from age-matched sibling controls based on pigmentation[69]. Single larvae are distributed into individual wells of flat-bottomed 96-well microplates (MultiScreen 96-well Transport Receiver

Plate, Millipore, Billerica, MA) in a volume of 100 μL of E3 medium per well and the microplates are placed inside the imaging chamber. 2x working stocks of each compound are prepared in E3 medium and the DMSO concentration is adjusted to 2%. 100 μL of the 2x working stock is added to each well ($n = 10$ larvae per experimental condition), resulting in the indicated screening concentrations and a final DMSO concentration of 1%. Four hours after compound addition, a 30 min locomotor activity recording is acquired.

The positions of all larvae are automatically detected in each frame using a custom MATLAB tracking algorithm that we developed previously[33]. Briefly, the algorithm generates a binary image of all larvae and performs a skeletonization step. The tip of the head and the tip of the tail are automatically identified and five equally spaced vertex points are positioned along the detected centerline of each larva. These five points are used to calculate six distinct behavioral features. The two anterior-most points are used to define the body centerline and calculate swimming velocity metrics while the three posterior-most points are used to define the tail centerline and calculate tail-specific metrics. The mean forward swimming velocity ($FV_{mean}$; pixels s$^{-1}$) is calculated based on the distance traveled by the larva parallel to the body centerline. The mean angular swimming velocity ($AV_{mean}$; pixel s$^{-1}$) and the standard deviation (SD) of the angular swimming velocity ($AV_{STD}$; pixels s$^{-1}$) are calculated based on the distance traveled by the larva perpendicular to the body centerline. We set a high velocity swimming threshold (values greater than $FV_{mean} + FV_{STD}$) and use this threshold to calculate the percentage of time spent at high swimming velocity (HV%). The tail angle is calculated for each tail vertex point and the absolute values of all points are summed together. These values are used to calculate the mean tail bending angle ($TB_{mean}$; degrees). Mean change in tail angle ($dTB_{mean}$; degrees s$^{-1}$) is based on the sum of change in tail angle for each tail vertex point. For each behavioral feature, we calculate the mean of all 10 larvae and normalize this value to the untreated (1% DMSO) mutant group (see Supplementary Table 3, Behavioral Metrics columns).

In order to calculate a combined behavioral score, we first determine how much each behavioral feature diverges from its mean baseline value in the untreated mutant group (see Supplementary Table 3, Divergence from Baseline columns). We then define the combined behavioral score ($x_{combined}$) as the square root of the arithmetic mean of the squares of each of the six divergence values.

The combined behavioral score represents the root mean square of the divergences and therefore serves as a measure of the magnitude of behavioral changes that are triggered by a given compound or duotherapy combination relative to baseline behavior in untreated larvae (see Supplementary Table 3, Side Effect column). The standard deviation of the combined behavioral score is obtained by first calculating the divergence of each behavioral feature for each of the 10 individual larvae in the condition of interest. A combined behavioral score is calculated for each individual larva and the standard deviation of all 10 individual scores is determined.

**LFP recording and analysis**. At 5 dpf, larvae are embedded in ultra-low gelling temperature agarose. At least five larvae are recorded for each experimental condition. Individual larvae are placed in a recording chamber with 50 mL of E3 solution and a 45 min pre-exposure LFP recording is obtained. After the baseline recording is complete, 50 mL of a 2x working stock of the test compound (s) is added to the recording chamber. Following compound administration, a second LFP recording is made 4+ h post-exposure. Recording electrodes are made by pulling a 1 mm outer diameter capillary (BF100–78–10, Sutter Instrument Company, Novato, CA), which is filled with 1M chloride solution. A Ag/Cl wire (64–1320, Harvard Apparatus, Holliston, MA) is then placed inside the electrode, the electrode is connected to a preamplifier (RHD2216, Intan Technologies, Los Angeles, CA), and the preamplifier is connected to a low-power digital acquisition chip (RHD2000, Intan Technologies). The signal from acquisition board is recorded using Intan MATLAB GUI software (MATLAB 13, Mathworks, Natick, MA). The data are then analyzed using an automated seizure detection algorithm originally developed to analyze EEG signals[46], which we have adapted to zebrafish LFP recordings. The algorithm uses higher order statistical moments as features to classify electrophysiological signals into seizure and non-seizure classes. Higher order moments are extracted from intrinsic mode functions, which are obtained by adaptively decomposing the signal using the empirical mode decomposition (EMD) method. The automated seizure detection algorithm is trained to identify seizures using LFP recordings from scn1lab mutants exposed to light stimuli as a training dataset. We then use the algorithm to measure spontaneous seizure frequency in compound-treated scn1lab mutants. Baseline seizure frequency is determined for each larva based on the pre-exposure LFP recording. The post-exposure frequency is then normalized to the baseline frequency and an efficacy score is calculated (untreated mutants = 0; seizure-free larvae = 1.00; Supplementary Table 3).

**Reporting summary**. Further information on research design is available in the Nature Research Reporting Summary linked to this article.

## Data availability
The data that support the findings of this study are available from the corresponding author upon request. The source data underlying Figs. 2c, 3, and Supplementary Figs. 1, 4, 11, and 13 and Supplementary Tables 2, 3, and 5 are provided as a file. The atlas is available online from GitHub (https://github.com/rezaie99/NC-18-28140.git).

## Code availability
Core source code, including (1) Python source code for motion correction and timeline extraction of images, (2) Bash files to run external tools (ANT) for registration, (3) Python code for frequency and synchronization analysis, (4) Python code for LFP analysis and correlation, and (5) MATLAB code for cluster analysis and statistical calculation, is available online from GitHub (https://github.com/rezaie99/NC-18-28140.git).

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

## Acknowledgments

We thank the following funding sources: Packard Award in Science and Engineering, NIH Director's Pioneer Award (DP1-NS082101), Harvard/MIT Broad Institute's SPARC Award, NIH Transformative Research Award (R01 NS073127), Swiss Federal Institute of Technology (ETH) Zurich and a postdoctoral fellowship from the Epilepsy Foundation. This project also has received funding from the European Research Council (ERC) under the European Union's Horizon 2020 research and innovation program (grant agreement No 818179). All zebrafish used in these studies were raised and maintained in the Koch Institute Zebrafish Core Facility directed by Drs. Jacqueline A. Lees and Adam Amsterdam.

## Author contributions

M.G-R. and P.M.E. designed the experiments. M.G-R. developed the light-sheet platform and the LFP recording platform, performed all experiments, and carried out all data analysis. M.G.-R. and Y.W. developed the video tracking system and locomotor tracking algorithms. P.M.E and M.G.-R. prepared the manuscript. The principal investigator and supervisor M.F.Y. conceived the study and worked on the preparation and editing of the manuscript.

## Additional information

**Competing interests:** The authors declare no competing interests.

