## [Peer Review File · Nature Communications]

Reviewers' Comments:

Reviewer #1:

Remarks to the Author:

I have reviewed this manuscript before for another journal. In the review I gave the authors extensive feedback, after which they made the suggested changes to improve the manuscript. These changes are also reflected in the current submission. I have no further comments to add, and remain enthusiastic.

Reviewer #2:

Remarks to the Author:

In this manuscript, Ghannad-Rezaie and colleagues describe a novel high-throughput light-sheet imaging setup for larval zebrafish, that incorporates automated animal handling, drug delivery, and multiple rounds of light-sheet Ca²⁺ imaging. They use this powerful system to interrogate the aberrant inter-area functional connectivity observed in an epileptic zebrafish mutant model of Dravet syndrome, and screened a panel of neuromodulatory drugs in search of those that can correct these functional connectivity deficits. Finally, they show that by combining drugs that are able to correct functional deficits, but do so in different ways, they are able to identify polytherapy regimens that outperform single drugs when scoring behavioural abnormalities, the presence of epileptic events, and functional connectivity. This is a technically impressive study, with potentially important implications for the rational design of polytherapy strategies, and the potential advantages of using larval zebrafish brain activity in a high-throughput system as a more direct and practical readout of the neural effect of therapeutic compounds and their combinations.

My concerns about this paper are as follows:

1) The main message of this paper appears to be that by directly measuring altered functional connectivity induced by drugs, and using these unique patterns to select individual drugs to combine, one can do a better job at selecting polytherapy combinations than if one had just selected two drugs based on either behavioural data, or based on known and complementary mechanisms of action. While I think this is a rational argument and potentially powerful strategy, I don't believe that they have directly demonstrated this. They show that the combinations outperform any individual drug, but this is not necessarily surprising since they were selected for being beneficial and are thought to have different mechanisms of action. I believe this point would be made much stronger if the authors could demonstrate that selecting compounds in a manner independent of their functional connectivity analyses (e.g. the top hits for rescuing behavioural abnormalities), does not result in as successful a combination as a Fluoxetine + Mifepristone.

2) The description of the anatomical analyses is insufficient. They segmented their reference brain into 55 areas, but I can not find any information about what these areas are (they don't seem to be listed in the methods) and what their segmentation actually looks like. While perhaps it isn't central to the message of the paper, I think having a more detailed description of the areas will be important when readers may try to understand the described functional associations. Ideally, I would encourage the authors to make the segmentation available for download, so that readers can understand the 3D relationships among all of the regions and how they are defined. I also note that in Fig 4, where the authors have marked the Thalamus does not seem correct, and instead I think this should be the Hypothalamus.

3) There is no description of how accurate the anatomical alignments are after registration. From the

methods, the authors use an affine transformation for alignment. Most recent approaches in zebrafish have used non-linear/warping registration, and so it is important to understand how reliable this registration method is, and by consequence how reliably each neuron can be assigned to an anatomical region. I feel this may be especially important because of the large z-step sizes employed here (20 or 40 μ M), which may in fact be larger than some larval zebrafish brain regions, depending on the granularity of the segmentation.

4) The authors describe looking for adverse "side effects" by measuring compound-induced behavioural alterations in mutant animals. This is confusing to me, as I understood that the mutants showed aberrant behaviour. Thus it is unclear to me if this experiment speaks more towards "side-effects", or rather correcting mutant pathology, which I expect would be desirable. The rationale here needs more explanation. Would it not make more sense to look for a lack of effects in wild-type animals, rather than a lack of effects in the mutants that the drugs are aiming to correct?

5) For the analyses of functional connectivity, the authors first extract super-voxels, which they claim mostly reflect individual neurons based on their \sim 7 μ m average size. Is it not possible to observe some of these voxels relative to the anatomy to determine what they tend to identify? Also, the analysis performs correlational analyses for all pairs of active supervoxels in pairs of regions. Is this analysis superior to simply correlating the average fluorescence traces in each area? I think the authors should explain why they chose this analysis, and advantages/disadvantages over a more simple global pooling of signals in each area. The authors also refer to calculating "cross-correlations". It does not seem to me like they are measuring the cross-correlation between the two signals (at least how I understand it), but rather simply the correlation coefficient. I think this should either be changed to "correlation", or a more detailed explanation of how "cross-correlation" is used is required.

6) In the introduction – line 118, the authors state they implemented "a high speed light-sheet microscope that can image the brain at single-cell resolution in 50 ms". I think this is an overstatement, because they are imaging in 40 μ m steps, which is about 10x the size of a zebrafish neuron. Therefore, while they presumably can image at cellular resolution at \sim 2 μ m step sizes, with the current imaging protocol they do not image all neurons. The authors do acknowledge this in the Methods section, estimating they detect \sim 15% of neurons. Therefore, I think such claims of brain-wide and cellular resolution should be toned down.

Reviewer #3:

Remarks to the Author:

The authors present evaluation of the use of connectivity fingerprints of activated brain networks in zebrafish larvae to find novel duo-therapeutic combinations for the treatment of epilepsy. The approach is described as a promising alternative to a behavioral or target based-approach of drug discovery in the field of neurological and psychiatric diseases. Overall, the manuscript is well written, and adds to the description of (dis)functional connectivity in a pathological condition of the zebrafish brain. Moreover, also the technical approach (high-speed light sheet microscopy, transgenic GCaMP5G zebrafish with Dravet background, connectivity calculations) is well described, sound and advanced.

However, due to unconvincing pharmacological data, it is doubtful whether the constructive and predictive validity of the platform is high. The authors fail to mention that Dravet patients are highly resistant to antiseizure medication and that some drugs can even aggravate the situation. For instance, sodium channel agents-including carbamazepine should be strictly avoided to treat Dravet patients, a compound that ranked very high in the connectivity fingerprint (Suppl Table 2). So if single treatment conditions do not result in an outcome that is in agreement with clear clinical data, one can

wonder what the relevance (validity) is of the duo-therapeutic approach. Moreover, although changes in functional connectivity are also present disorders like Alzheimer's disease, autism, schizophrenia, depression, etc., it is unrealistic to think that all these human diseases can be modeled in zebrafish larvae. So also here the expectations are somewhat overrated.

Reviewer #4:

Remarks to the Author:

This is an outstanding study in which the authors employed an innovative approach to identify synergistic neuromodulators that normalize brain function in a zebrafish model of Dravet Syndrome, a catastrophic, drug-resistant, epileptic syndrome. Although zebrafish larvae have been used in several screens to identify novel neuromodulators (and indeed a zebrafish Dravet Syndrome model has already been successfully used as the basis for such a screen) these previous efforts typically screen single compounds while assaying simple locomotor behavior of field potential recordings. This study moved beyond these approaches, which may not capture differences in complex internal states, by developing a high-throughput means to capture the functional connectivity of 55 pairs of brain regions. The functional connectivity 'fingerprint' of Dravet Syndrome is distinct from wild-type zebrafish animals, as revealed by cluster analysis, and is modulated in different ways when treated with compounds from panel of 24. Compounds with complementary fingerprints were rationally selected for combined application, and this polytherapy approach was able to restore brain connectivity more effectively, and with fewer toxic effects, than any single compound tested.

This study is thorough, multi-faceted, convincing and of profound interest in at least two capacities. The first is the identification of a possible new therapeutic strategy to treat Dravet Syndrome. The two most efficacious drugs, fluoxetine and mifepristone, have not been widely used to treat epilepsies, so this study highlights these compounds for further investigation. The second is the novelty of the approach that the authors employed, which sidesteps a substantial challenge in combinatorial drug screens. The author's platform is ripe for future screens, using other disease models and/or other small molecule libraries, and this manuscript is likely to shape thinking about polytherapy drug screening strategies. This paper is well-written, the statistical analysis and Methods section seem appropriate, and I have no concerns. I do have a few thoughts that the authors may want to consider addressing:

-A zebrafish model of Dravet Syndrome has already been used to screen small molecule library, which has identified Clemizole and other compounds that modulate serotonin signaling as a promising therapy (Grittin et al., 2017; Sourbron et al., 2016; Dinday and Baraban, 2015). Is there any way to compare the efficacy of fluoxetine/mifepristone to these previous studies? Were common compounds tested to compare results?

-It might be useful to provide videos of the behavioral change after application of fluoxetine/mifepristone. The change in kinematic parameters, shown in Supplementary Figure 9, is impressive but videos can often provide a more intuitive understanding.

-Text in some of the figures is so small as to be difficult to read, for example the drugs in 2C and the text Supplementary Figure 9. In addition, it would be useful, although not completely necessary, to label the axes in each of the connectivity fingerprint figures (2C, Supplementary Figure 6, and Supplementary Figure 7).

We would like to thank all the reviewers for their comments. Here we provided point by point answer to all the comments.

Reviewer #1 (Remarks to the Author):

I have reviewed this manuscript before for another journal. In the review I gave the authors extensive feedback, after which they made the suggested changes to improve the manuscript. These changes are also reflected in the current submission. I have no further comments to add, and remain enthusiastic.

We want to thank the reviewer for the constructive comments that helped us improve the manuscript significantly.

Reviewer #2 (Remarks to the Author):

In this manuscript, Ghannad-Rezaie and colleagues describe a novel high-throughput light-sheet imaging setup for larval zebrafish, that incorporates automated animal handling, drug delivery, and multiple rounds of light-sheet Ca²⁺ imaging. They use this powerful system to interrogate the aberrant inter-area functional connectivity observed in an epileptic zebrafish mutant model of Dravet syndrome, and screened a panel of neuromodulatory drugs in search of those that can correct these functional connectivity deficits. Finally, they show that by combining drugs that are able to correct functional deficits, but do so in different ways, they are able to identify polytherapy regimens that outperform single drugs when scoring behavioural abnormalities, the presence of epileptic events, and functional connectivity. This is a technically impressive study, with potentially important implications for the rational design of polytherapy strategies, and the potential advantages of using larval zebrafish brain activity in a high-throughput system as a more direct and practical readout of the neural effect of therapeutic compounds and their combinations.

We thank the reviewer for his/her very helpful comments. Below we addressed his/her concerns in detail and modified the manuscript accordingly.

My concerns about this paper are as follows:

- 1) The main message of this paper appears to be that by directly measuring altered functional connectivity induced by drugs, and using these unique

patterns to select individual drugs to combine, one can do a better job at selecting polytherapy combinations than if one had just selected two drugs based on either behavioural data, or based on known and complementary mechanisms of action. While I think this is a rational argument and potentially powerful strategy, I don't believe that they have directly demonstrated this. They show that the combinations outperform any individual drug, but this is not necessarily surprising since they were selected for being beneficial and are thought to have different mechanisms of action. I believe this point would be made much stronger if the authors could demonstrate that selecting compounds in a manner independent of their functional connectivity analyses (e.g. the top hits for rescuing behavioural abnormalities), does not result in as successful a combination as a Fluoxetine + Mifepristone.

Picking compounds based only on simple behavioral readouts (such as locomotor activity, etc.) leads to numerous false positives, as has been shown in previous publications from our lab (Eimon et al. 2018, Nature Communications) and others (Baraban et al. 2013, Nature Communications). This is presumably due to that fact that drugs which are sedatives, paralytics, or broadly impair CNS function can all give the appearance of reducing seizure activity. Indeed, of the 31 drugs which appeared to be "hits" in our original behavioral screen, only 6 showed strong anti-seizure activity on their own when assessed using LFP. Therefore, picking combinations of compounds based on simple behavioral readouts is unlikely to be a successful strategy.

In contrast, we agree that picking combinations based on differing mechanisms of action (MOA) may indeed be a viable approach in some cases. However, this requires detailed knowledge about all compounds in the screen, including primary MOA, off-target activity, metabolites, etc. Our approach can be used to classify even compounds with entirely unknown MOAs. We therefore view our technology as a superior tool that can be used to classify compounds and choose combinations based on their actual in vivo biological activities and effects on circuits. This offers researchers a powerful new classification tool that is in many respects complimentary to a more traditional MOA-based approach.

Although not included previously, we have very strong behavioral evidence that only combining the top behavioral-hit compounds does not produce the synergy we observed with our approach. Selecting top behavioral hits with different mechanism of actions (MOAs) is a better approach, but even this

approach still does not achieve the high 90% efficacy we achieve with our functional connectivity analysis. We have now revised the text of our manuscript and added new supplementary data (**Supplemental Figure 12**) in order to further support our finding that selecting the top-hit compounds only based on behavioral metrics and MOAs is not a good polytherapy strategy and the clustering method presented in this paper is essential.

Based on behavioral assessments, we now included another drug combination. We examined a new combination of the top behavioral hits from our previous study (Eimon et al. 2018, Nature Communications) with different MOAs (without using our functional connectivity analysis) as suggested: allopregnanolone and progesterone (top hit with different MOAs; GABA agonist and progesterone receptor agonist).

With respect to the combination based on top behavioral hits, the fluoxetine (F) + mifepristone (M) combination based on our connectivity analysis still clearly gives the highest efficacies with the least side effects. For example, allopregnanolone (A; 100%) + progesterone (P;100%) gives 83% efficacy, which is not as good as mifepristone (30%) + fluoxetine (30%) (91% efficacy), and very importantly has a behavioral side effect score of 0.75 (compared to only 0.38 for the F+M combination). We now included the new data in **Supplemental Figure 12**. Clearly, just combining the top behavioral hits, even after selecting those with different known MOAs is not as good as the combination based on our connectivity analysis.

Supplemental Figure 12:

(A) The Top Combination of drugs based on Behavioral Hits with different Mechanism of Actions.

(B) The Top Combination of drugs based on Functional Connectivity Analysis

2) The description of the anatomical analyses is insufficient. They segmented their reference brain into 55 areas, but I can not find any information about what these areas are (they don't seem to be listed in the methods) and what their segmentation actually looks like. While perhaps it isn't central to the message of the paper, I think having a more detailed description of the areas will be important when readers may try to understand the described functional

associations. Ideally, I would encourage the authors to make the segmentation available for download, so that readers can understand the 3D relationships among all of the regions and how they are defined.

We have now clarified the manuscript to avoid any confusion: We only have 11 areas (not 55) in our segmentation which resulted in 55 connectivities between pairs of areas, which we used in our analysis. The list of areas used in the analysis were listed in Figure 4 and supplemental Figure 9 description. To make the manuscript clearer, we now added a new supplementary table (**Supplemental Table 5**) to list all the areas used in our analysis. We also made the segmentation available online. We have now uploaded our brain atlas segmentations to GitHub along with other codes (online from GitHub <https://github.com/rezaie99/NC-18-28140.git>).

Supplemental Table 5: Accuracy of brain segmentation. For each brain area in our analysis, we subtract the result of segmentation algorithm from the manual segmentation and then normalize the difference to the size of the manual segmentation.

Area Name	Area Symbol	Average Segmentation Error (%)
Cerebellum	Ce	4.2
Hindbrain left	HBl	4.6
Hindbrain right	HBr	4.7
Olfactory bulb left	OBl	6.3
Olfactory bulb right	OBr	6.4
Optical tectum, left	OTl	5.4
Optical tectum, right	OTr	5.6
Pallium	Pa	16.2
Sub-pallium	Spa	17.2
Habenula	Ha	6.4
Thalamus	Th	8.5

I also note that in Fig 4, where the authors have marked the Thalamus does not seem correct, and instead I think this should be the Hypothalamus. The Fig. 4 is just for visual representation and not anatomically accurate. We clarified this now in the figure caption. We also modified the schematic in Fig. 4 to better represent of the anatomical location of Thalamus.

3) There is no description of how accurate the anatomical alignments are after registration. From the methods, the authors use an affine transformation for alignment. Most recent approaches in zebrafish have used non-linear/warping registration, and so it is important to understand how reliable this registration method is, and by consequence how reliably each neuron can be assigned to an anatomical region. I feel this may be especially important because of the large z-step sizes employed here (20 or 40uM), which may in fact be larger than some larval zebrafish brain regions, depending on the granularity of the segmentation.

We think this concern has raised from the assumption that we used large number of areas (55 areas) in our analysis. As noted in response to the previous comment, we use only 11 relatively large brain areas in our segmentation, and therefore our clustering analysis is not very sensitive to the segmentation accuracy. We now added a supplemental table (see **Supplemental Table 5** above) that lists all the areas used in our analysis. To better quantify our segmentation accuracy, we also included the average segmentation error in this table (as compared to manual segmentation).

To address the reviewer's concern about the accuracy of the segmentation, we estimated the effect of the segmentation accuracy on our clustering algorithm. With the 20um slice thickness we use in our anatomical imaging, the affine transformation we used for segmentation is accurate enough for our analysis. To demonstrate our segmentation is sufficiently accurate, we calculated the clusters (C1-C3; **Fig. 2**) after randomly modifying the segmentation such that the new segmentation has only 95% overlap with the original segmentation and therefore adding 5% simulated error to the segmentation results. As you can see, C1, C2 and C3 clusters are still identical to the original clusters in the figure 2C. We repeated this method with 10%, 20% and 30% segmentation error and measured the change in the clustering results. We measured the distance between original C1 and C2 clusters (BCD) and within C1 cluster distance (WCD). Our results show more than 20% segmentation error in average affects the outcome of our clustering algorithm, while less than 10% average segmentation error does not affect our clustering at all. We added **Supplemental Figure 11** to the manuscript to clarify this point.

Supplemental Figure 11: Segmentation algorithm is sufficiently accurate for the clustering algorithm. We calculated the clusters after randomly

modifying the segmentation. We then measured two metrics for each clustering results: The average distance between original C1 and C2 clusters (BCD) and the average distance within C1 cluster (WCD). (A) Calculated BCD and WCD for original segmentation, segmentation with 5%, 10%, 20% and 30% error. (B) Original clustering results, (C) clustering results with 10% clustering error and (D) 20% segmentation noise. (E) The average distance between original C1 and C3

(A) BCD (C1-C2) and WCD (C1)

(C) 10% Noise

(D) 20% Noise

(E) BCD (C1-C3) and WCD (C1)

4) The authors describe looking for adverse “side effects” by measuring compound-induced behavioural alterations in mutant animals. This is confusing to me, as I understood that the mutants showed aberrant behaviour. Thus it is unclear to me if this experiment speaks more towards “side-effects”, or rather correcting mutant pathology, which I expect would be desirable. The rationale here needs more explanation. Would it not make more sense to look for a lack of effects in wild-type animals, rather than a lack of effects in the mutants that the drugs are aiming to correct?

It would not be accurate to test the drug side effects on wild types. For instance, consider a perfect drug (say an antiepileptic) restoring E/I balance in an idealized seizure model that has only E/I balance deficit. The same drug would disturb the E/I balance in an otherwise normal brain (i.e. wild type) leading to all kinds of effects that would not be present in the seizure model under drug, which could be mistaken as drug’s side effect.

We consider side effects as the collection of new abnormal behavioral metrics that arise after treatment of mutants with the drug. We choose these side-effect behavioral metrics not to overlap with the behavioral metrics that represent/define the abnormal behaviors of mutants. These later metrics are used to calculate only the efficacy of the drugs.

Because the side effect score is calculated from a collection of different (and often incomparable) behavioral metrics, there is some ambiguity in weighting the importance of different behavioral metrics to generate a single side effect

metric (A clinical example to this challenge would be weighting hairloss vs. skin rash vs. nausea vs. diarrhea vs. headache side effects to create a single metric representing all). This is why we provided how we calculated the side effect metric where we equally weighted all side effects to prevent ourselves from biasing the results. In our analysis, we look at the changes in the behavior during three spontaneous activities and the two phases of stimuli driven behavior of the animal. The side effect score is based on six distinct behavioral metrics quantified over a 30-minute interval spontaneous activity.

5) For the analyses of functional connectivity, the authors first extract supervoxels, which they claim mostly reflect individual neurons based on their ~7 μ m average size. Is it not possible to observe some of these voxels relative to the anatomy to determine what they tend to identify?

HuC promoter is used widely in Zebrafish literature to drive pan-neuronal expression of GCamp (Ahrens et al. Nature Method 2013). Therefore, we are very confident that the active pixels with significant changes in $\Delta F/F$ are reflecting the calcium signal from neurons. A supervoxel usually represents a single neuron. However, it is possible that our algorithm identifies a cluster of synchronous neurons in close proximity as a supervoxel. Since we use correlational analysis at the end, these occasional clusters do not affect the ultimate results. In **Supplemental Figure 13**, we showed how much the voxel resolution affects the clustering result.

Also, the analysis performs correlational analyses for all pairs of active supervoxels in pairs of regions. Is this analysis superior to simply correlating the average fluorescence traces in each area?

I think the authors should explain why they chose this analysis, and advantages/disadvantages over a more simple global pooling of signals in each area.

Here, we demonstrate that the correlational analysis of active supervoxel pairs outperforms a simpler global pooling. First, we calculated the correlation coefficients of global pooling as suggested and then used the results for compound clustering. To compare the quality of clusters, we measured two metrics: the distance between wild type cluster and mutant cluster (BCD, larger is better), and the distance within the wild type cluster (WCD, smaller is better). Our results show that both metrics deteriorate when we use global pooling instead of voxel correlation: BCD decrease ~56% (from 1.62 to 0.91)

and WCD increase ~110% (0.38 to 0.80). Furthermore, we demonstrate that BCD/WCD ratio decreases for larger voxels (**Supplemental Figure 13**).

There are three reasons that our supervoxel correlation method have better signal to noise ratio and therefore outperformed simple pooling:

- 1- **Background Noise:** Most neuron pairs are not connected, therefore the correlation for these pairs should be zero. However, since we are calculating the correlation in a relatively short time scale, the correlation between the calcium signals is not exactly zero. This non-zero correlation introduces a background noise into the correlation calculations. Non-active neurons and the void between neurons also contribute to this background noise.
- 2- **Negative and Positive Correlations:** The correlation between the connected neuron pairs can be positive(excitation) or negative (inhibition). In the aggregate signal positive and negative correlation cancels out. In our method we average the absolute value of correlations to avoid this issue.
- 3- **The Voxel Size:** The size of the voxel is not a contributing factor in our analysis. However, if one instead averages the fluorescence intensity over a volume, both the bigger cells and the cells within focus are overrepresented in the average intensity of the area. The voxel preprocessing considers all cells equal regardless of size and the focus plane.

Supplemental Figure 13. Voxel size effects the clustering algorithm results. To compare quality of clusters, we measured two metrics: the distance between wild type cluster and mutant cluster (BCD, larger is better), and the distance within wild type cluster (WCD, smaller is better). Our results show both metrics degraded when we use global pooling instead of voxel correlation: BCD decrease ~56% (from 1.62 to 0.91) and WCD increase ~110% (0.38 to 0.80). Furthermore, we increased the size of our voxels (in average 14 μ m, 28 μ m, and 56 μ m) to demonstrate more gradual degradation of clusters as the voxels get bigger and eventually each area is just one voxel.

BCD (Mutant- wild type) and WCD (wild type)

The authors also refer to calculating “cross-correlations”. It does not seem to me like they are measuring the cross-correlation between the two signals (at least how I understand it), but rather simply the correlation coefficient. I think this should either be changed to “correlation”, or a more detailed explanation of how “cross-correlation” is used is required.

We also change the term “cross-correlation” to “correlation” and “absolute correlation coefficient” for clarity.

6) In the introduction – line 118, the authors state they implemented “a high speed light-sheet microscope that can image the brain at single-cell resolution in 50 ms”. I think this is an overstatement, because they are imaging in 40 μ m steps, which is about 10x the size of a zebrafish neuron. Therefore, while they presumably can image at cellular resolution at ~2 μ m step sizes, with the current imaging protocol they do not image all neurons. The authors do acknowledge this in the Methods section, estimating they detect ~15% of neurons. Therefore, I think such claims of brain-wide and cellular resolution

should be toned down.

We thank the reviewer for the comment. We now refer to our imaging technique as 'image ~15% of the brain', rather than 'whole brain imaging' to be more accurate. We also like to point out that previous pioneering studies using light-sheet imaging of zebrafish brain also claimed whole-brain cellular-resolution keywords when describing their functional imaging under similar conditions (Ahrens et al. Nature Method 2013).

Reviewer #3 (Remarks to the Author):

The authors present evaluation of the use of connectivity fingerprints of activated brain networks in zebrafish larvae to find novel duo-therapeutic combinations for the treatment of epilepsy. The approach is described as a promising alternative to a behavioral or target based-approach of drug discovery in the field of neurological and psychiatric diseases. Overall, the manuscript is well written, and adds to the description of (dis)functional connectivity in a pathological condition of the zebrafish brain. Moreover, also the technical approach (high-speed light sheet microscopy, transgenic GCaMP5G zebrafish with Dravet background, connectivity calculations) is well described, sound and advanced.

However, due to unconvincing pharmacological data, it is doubtful whether the constructive and predictive validity of the platform is high. The authors fail to mention that Dravet patients are highly resistant to antiseizure medication and that some drugs can even aggravate the situation. For instance, sodium channel agents-including carbamazepine should be strictly avoided to treat Dravet patients, a compound that ranked very high in the connectivity fingerprint (Suppl Table 2). So if single treatment conditions do not result in an outcome that is in agreement with clear clinical data, one can wonder what the relevance (validity) is of the duo-therapeutic approach. Moreover, although changes in functional connectivity are also present disorders like Alzheimer's disease, autism, schizophrenia, depression, etc., it is unrealistic to think that all these human diseases can be modeled in zebrafish larvae. So also here the expectations are somewhat overrated.

We thank the reviewer for his/her comments. The reviewer is concerned that carbamazepine is dangerous for Dravet patients but the zebrafish connectivity fingerprint shows it is high ranking, hence questioning the value of our approach. Below, (1) we first explain why we would NOT pick carbamazepine for any cocktail (even solely based on the zebrafish Dravet model). Next, (2)

we explain broader implications of our method, and why its potential is not limited to the validity of any particular disorder/disease/animal model.

(1) The functional connectivity alone does not imply that a drug should be included in final cocktail. Carbamazepine ranks highly in our clusters because it indeed normalizes functional connectivity. However, in our previous paper which combines deep behavioral phenotyping with high-throughput LFP analysis, we clearly show that Carbamazepine performs badly on seizure scores (Eimon et. al. 2018, Nature Communications). In this study, since our emphasis was on showing the power of functional connectivity, we only used simple behavioral metrics but have not included our LFP analysis to keep other aspects of this study simple and more transparent.

Looking in more detail at the specific example the reviewer raises (carbamazepine), we note that one obvious difference between SCN1A phenotypes in zebrafish and mammals is that seizures arise from haploinsufficiency in mammals but require the loss of both copies of the gene in fish. The reasons for this difference have not been fully established, but might reflect well-characterized evolutionary differences in the repertoire of voltage-gated sodium ion channels present in teleost fish and tetrapods (see Widmark et al., 2011, "Differential Evolution of Voltage-Gated Sodium Channels in Tetrapods and Teleost Fishes," *Mol. Biol. Evol.* 28:859–871). Because human patients with Dravet syndrome retain one functional copy of SCN1A, they presumably still express functional protein in inhibitory interneurons, although not at sufficient levels to fully protect against seizures. It is therefore not surprising that patients are especially sensitive to sodium channel blockers such as carbamazepine, as these drugs have the effect of further diminishing the already reduced activity of SNC1A and consequently aggravating the phenotype. In contrast, the fact that seizure phenotypes in zebrafish require the loss of both copies of *scn1lab* (the SCN1A ortholog) might mean that voltage gated sodium ion channel activity in inhibitory interneurons is already effectively abolished. As such, drugs like carbamazepine cannot further aggravate seizures and may in fact help to reduce them by blocking sodium channels in excitatory neurons (as they do in human patients with most other forms of epilepsy).

(2) The central point of our study is that the distinctive alteration of functional connectivities induced by a drug is a good marker for selecting drug combinations for polytherapy, and it can outperform drugs/cocktails that are selected solely based on the mechanisms of actions (MOAs) and/or

behavioral outcomes. While we acknowledge that zebrafish models likely never recapitulate every feature of a given disorder in humans, we believe our approach still has significant potential because it is translatable to many animal models and may even be directly applied in the clinic by fMRI/EEG/EECoG to discover polytherapy cocktails. In fact, our lab recently received one of the largest single-PI grants of European Union (ERC Consolidator) to further develop and test our approach on rodents (based on our preliminary data in zebrafish and rats) with minimally invasive functional connectivity measurements that will also be translatable to primates (including humans). We will be happy to provide the editor our proposal. Even in situations where drugs produce a different biological effect in zebrafish than mammals, simply knowing that a given cluster of drugs all produce SIMILAR effects in zebrafish may point researchers toward critical (and previously unsuspected) commonalities in their underlying mechanisms of action.

We edited the manuscript to make these points clearer.

Reviewer #4 (Remarks to the Author):

This is an outstanding study in which the authors employed an innovative approach to identify synergistic neuromodulators that normalize brain function in a zebrafish model of Dravet Syndrome, a catastrophic, drug-resistant, epileptic syndrome. Although zebrafish larvae have been used in several screens to identify novel neuromodulators (and indeed a zebrafish Dravet Syndrome model has already been successfully used as the basis for such a screen) these previous efforts typically screen single compounds while assaying simple locomotor behavior or field potential recordings. This study moved beyond these approaches, which may not capture differences in complex internal states, by developing a high-throughput means to capture the functional connectivity of 55 pairs of brain regions. The functional connectivity 'fingerprint' of Dravet Syndrome is distinct from wild-type zebrafish animals, as revealed by cluster analysis, and is modulated in different ways when treated with compounds from panel of 24. Compounds with complementary fingerprints were rationally selected for combined application, and this polytherapy approach was able to restore brain connectivity more effectively, and with fewer toxic effects, than any single compound tested.

This study is thorough, multi-faceted, convincing and of profound interest in at

least two capacities. The first is the identification of a possible new therapeutic strategy to treat Dravet Syndrome. The two most efficacious drugs, fluoxetine and mifepristone, have not been widely used to treat epilepsies, so this study highlights these compounds for further investigation. The second is the novelty of the approach that the authors employed, which sidesteps a substantial challenge in combinatorial drug screens. The author's platform is ripe for future screens, using other disease models and/or other small molecule libraries, and this manuscript is likely to shape thinking about polytherapy drug screening strategies. This paper is well-written, the statistical analysis and Methods section seem appropriate, and I have no concerns. I do have a few thoughts that the authors may want to consider addressing:

We thank the reviewer for his/her concerns. Below we address all of them in detail.

-A zebrafish model of Dravet Syndrome has already been used to screen small molecule library, which has identified Clemizole and other compounds that modulate serotonin signaling as a promising therapy (Grittin et al., 2017; Sourbron et al., 2016; Dinday and Baraban, 2015). Is there any way to compare the efficacy of fluoxetine/mifepristone to these previous studies? Were common compounds tested to compare results?

In our previous work (Eimon et al. 2018, Nature Communications), we demonstrated that although these compounds do indeed reduce electrographic seizures at these concentrations, they also produce significant side effects. Previously all these drugs have been tested only at substantially higher concentrations than in our studies, which cause significant side effects that have been underestimated in these studies. For example, in Baraban et al. (2013, 2015), both drugs are used at a concentration of 1 mM in ~7% DMSO. In Griffin et al. (2017), they are tested at 250 μ M in the *scn1laa* mutant line (rather than the *scn1lab* line, which is used in our experiments). In contrast, our preliminary screen and subsequent LFP in [3] and behavioral analysis were all done using stiripentol at 20 μ M and diazepam at 10 μ M (with a final DMSO concentration of 1%). As noted in our previous manuscript, these concentrations were chosen based on the results of a preliminary touch-response assay in which larvae were initially exposed to all test compounds at 100 μ M and then to increasingly lower concentrations if their response appeared significantly abnormal. In order to confirm that discrepancies between our results and previous publications are due to concentration differences, we have retested stiripentol and diazepam at 100 and 200 μ M in our assays. We find that although spontaneous seizure-like events are indeed

reduced in LFP recordings from larvae exposed to higher concentrations, locomotor behaviors are abnormal, suggesting that the anti-seizure activity at such high concentrations is accompanied by profound side effects (sedation or other cognitive impairments).

Effect of stiripentol and diazepam at high concentrations on wild-type locomotor activity. Box-and-whisker plots showing mean swimming velocity (arbitrary units) for wild-type sibling controls 4-hours post-exposure to 1% DMSO alone or stiripentol or diazepam at the indicated concentrations (in 1% DMSO). Larvae were recorded in a 96-well plate using an automated tracking platform and 6 larvae were used per condition. Light stimuli were applied as described in Fig. 1a. Each light stimulus consists of two consecutive 500 millisecond light pulses separated by 1 second of darkness. Mean swimming velocities were calculated over the full 10-minute recording session. Tops and bottoms of each box represent the 1st and 3rd quartiles. Whiskers are drawn from the ends of the interquartile ranges (IQR) to the outermost data point that falls within ± 1.5 times the IQR. The line in the middle of each box is the sample median. Statistical significance was determined by Welch's t-test.

Although both diazepam (a Benzodiazepine) and stiripentol (a structurally novel modulator of the GABAA receptor) failed to improve brain activity patterns in our hands, previous publications suggest that they are capable of reducing spontaneous electrographic seizures in *scn1lab* mutant zebrafish. However, both compounds have only been evaluated at extremely high

concentrations (1 mM) in zebrafish. In contrast, we screened them at substantially lower levels (10-20 μM) based on results from our preliminary toxicity assessment, which revealed a noticeable reduction in touch response at concentrations as low as 100 μM . This suggests that higher concentrations may produce significant side effects in addition to anti-seizure activity. To determine if we could replicate published results using our platform and algorithms, we retested both diazepam and stiripentol at 100 and 200 μM . Under these conditions, we do indeed observe a significant reduction in the number of spontaneous seizure-like events in mutant larvae, however this is not accompanied by a corresponding improvement in the side effect. Additionally, when wild-type sibling controls are exposed to diazepam and stiripentol at these higher concentrations, we observe both a significant behavioral side effects. Based on these data, we conclude that the antiepileptic activity of Diazepam and stiripentol in *scn1lab* mutant larvae occurs only at concentrations that cause considerable off-target side effects. In the case of stiripentol, which has received orphan drug status for the treatment of DS, it should be noted that it is often combined with other AEDs clinically. At least some of its therapeutic activity is thought to arise from inhibition of their metabolism rather than from its own activity at the GABAA receptor⁵². Such combinatorial activity is not studied in our screen, which only uses individual compounds”

Similarly, with regards to clemizole, we note that in Griffin et al. (2017) it was tested at concentrations ranging from 30 to 400 μM with the following result: “Clemizole (Fig. 1A) exhibited antiepileptic activity at 300 and 400 μM (30-min exposure) and at 100 μM (90-min exposure)” (p. 673). These results are not inconsistent with our finding that clemizole at a much lower concentration (10 μM) fails to reduce seizure-like locomotor activity below our assay’s toxicity threshold of 50%. Indeed, even at this low concentration clemizole actually comes close to meeting our threshold at the 4-hour post-exposure time point. As with Diazepam and stiripentol, we have subsequently retested clemizole at 100 and 200 μM and find that it does indeed reduce spontaneous seizures in *scn1lab* mutants at higher concentrations. However, there are again clear indications that larvae experience significant side effects in addition to the desired anti-seizure activity at these concentrations. We added more reference to our previous work to clarify this point in the manuscript.

-It might be useful to provide videos of the behavioral change after application of fluoxetine/mifepristone. The change in kinematic parameters, shown in Supplementary Figure 9, is impressive but videos can often provide a more

intuitive understanding.

We now added a new video of the behavioral change after application of fluoxetine/mifepristone (**Supplemental Video 3**)

-Text in some of the figures is so small as to be difficult to read, for example the drugs in 2C and the text Supplementary Figure 9. In addition, it would be useful, although not completely necessary, to label the axes in each of the connectivity fingerprint figures (2C, Supplementary Figure 6, and Supplementary Figure 7).

We now provide high resolution version of Figure 2C as supplementary figure (**Supplemental Figure 11**). We also added labels to all axes of the connectivity fingerprint figures.

Reviewers' Comments:

Reviewer #2:

Remarks to the Author:

The authors have made a substantial effort to address all of my previous concerns, which I find satisfactory.

As a minor note - I tried to download the anatomical segmentation from the github side (https://github.com/rezaie99/NC-18-28140/tree/initial_commit/Atlas), but I was unable to identify the relevant file pertaining to the 11 regions. There is an 'atlas.img' file, which I assume contains the data, but this was not mountable on my system (Windows 10). I'd suggest that the authors increase the descriptions on the github site to make it clear where and how files can be accessed.

I have no further concerns.

Reviewer #3:

Remarks to the Author:

Regarding carbamazepine, the authors argue correctly that a functional connectivity analysis of a compound or cocktail should be complemented with other behavioral testing to confirm the pharmacological benefit. It is further stated that -in rare cases- a discrepancy might be observed between the two approaches. Unfortunately, a more thorough quantitative analysis of false positive results is lacking. This is a missed opportunity as the Dravet zebrafish model mimics well the clinical situation and hence all clinical tools are available to validate more accurately the method. In the absence of such data, the authors should adjust their expectations, and more critically discuss their outcome in anticipation of future results.

Reviewer #4:

Remarks to the Author:

I have no further comments. My concerns were thoroughly addressed. Really nice work.

We would like to thank the reviewers for the comments. Here we provided point by point answer to all the comments.

Reviewer #2 (Remarks to the Author):

The authors have made a substantial effort to address all of my previous concerns, which I find satisfactory.

As a minor note - I tried to download the anatomical segmentation from the github side (https://github.com/rezaie99/NC-18-28140/tree/initial_commit/Atlas), but I was unable to identify the relevant file pertaining to the 11 regions. There is an 'atlas.img' file, which I assume contains the data, but this was not mountable on my system (Windows 10). I'd suggest that the authors increase the descriptions on the github site to make it clear where and how files can be accessed.

I have no further concerns.

We would like to thank the reviewer for pointing out this problem. We now add a detail description to the github address to explain how to access the data in all platforms (MacOS, Windows and Linux.) We explained the software requirements for downloading the large files from github repository.

Reviewer #3 (Remarks to the Author):

Regarding carbamazepine, the authors argue correctly that a functional connectivity analysis of a compound or cocktail should be complemented with other behavioral testing to confirm the pharmacological benefit. It is further stated that -in rare cases- a discrepancy might be observed between the two approaches. Unfortunately, a more thorough quantitative analysis of false positive results is lacking. This is a missed opportunity as the Dravet zebrafish model mimics well the clinical situation and hence all clinical tools are available to validate more accurately the method. In the absence of such data, the authors should adjust their expectations, and more critically discuss their outcome in anticipation of future results.

We thank the reviewer for valuable feedback. We have completely characterized the effects of carbamazepine and all other chemicals we used on Dravet zebrafish model by deep-behavioral phenotyping, electrophysiology, and single-cell resolution entire-brain imaging in our current manuscript and in our last paper in Nature Communications (Eimon et. al. 2018).